# Emerging exotic compositional order on approaching low-temperature equilibrium glasses

Hua Tong [1,2] ✉ & Hajime Tanaka [2,3] ✉

The ultimate fate of a glass former upon cooling has been a fundamental problem in condensed matter physics and materials science since Kauzmann. Recently, this problem has been challenged by a model with an extraordinary glass-forming ability effectively free from crystallisation and phase separation, two well-known fates of most glass formers, combined with a particle-size swap method. Thus, this system is expected to approach the ideal glass state if it exists. However, we discover exotic compositional order as the coexistence of space-spanning network-like structures formed by small-large particle connections and patches formed by medium-size particles at low temperatures. Therefore, the glass transition is accompanied unexpectedly by exotic compositional ordering inaccessible through ordinary structural or thermodynamic characterisations. Such exotic compositional ordering is found to have an unusual impact on structural relaxation dynamics. Our study thus raises fundamental questions concerning the role of unconventional structural ordering in understanding glass transition.

When a liquid is cooled fast enough, it enters into a metastable supercooled state and finally solidifies into an amorphous solid, i.e., a glass, with a mixed character of mechanical rigidity and structural disorder[1,2]. This glass transition phenomenon represents a unique example of liquid-to-solid transitions without distinct structural changes, whose fundamental understanding remains premature despite considerable efforts over the years[3,4]. While an experimentally observed glass transition is a dynamic crossover where the diffusive relaxation freezes out on the experimental time scale[1], whether there exists an underlying thermodynamic phase transition or not has attracted considerable attention as a conceptually intriguing problem[5–9]. Such a link to the thermodynamic transition has been expected from the universality of dynamic and thermodynamic features of the glass transition[1,2]. The exploration of this problem crucially relies on the ability to approach the possible underlying phase transition and the knowledge of the unconventional structural ordering in glass formers upon slow cooling[10–13].

All simple materials crystallise upon cooling, forming periodically ordered lattice structures, a well-acknowledged fate of the low-temperature state of matter[14]. Therefore, in order to study the glass transition problem, we must introduce some frustration against crystallisation to the system to realise an excellent glass-forming ability, that is, the ability to maintain the metastability of the disordered liquid state upon cooling[14,15]. A common strategy is to use mixtures of different constituents[16,17]. In the case of computer simulations, constructed were standard models, such as the classic Wahnström and Kob-Andersen binary Lennard-Jones mixtures[18,19] and polydisperse systems with continuous particle-size distributions[16], representative of atomic and colloidal glasses, respectively. Invaluable advances in understanding glass transition have been made based on computational studies of these model glass formers over the past decades.

However, with the developments of computer technologies and simulation algorithms, one reaches deeper supercooling but confronts the instability of the liquid[20–22]. Besides direct crystallisation, phase separation has been observed in those standard models, which

[1]Department of Physics, University of Science and Technology of China, Hefei 230026, China. [2]Department of Fundamental Engineering, Institute of Industrial Science, University of Tokyo, 4-6-1 Komaba, Meguro-ku, Tokyo 153-8505, Japan. [3]Research Center for Advanced Science and Technology, University of Tokyo, 4-6-1 Komaba, Meguro-ku, Tokyo 153-8904, Japan. ✉e-mail: huatong@ustc.edu.cn; tanaka@iis.u-tokyo.ac.jp

proceed as demixing or fractionation of different kinds of particles in binary mixtures or polydisperse systems, respectively[20–27]. Although the phase separation is often followed by further crystallisation within each phase, e.g., the formation of face-centred cubic crystals in the pure large-particle phase in the Kob-Andersen model[21,23], it can be considered as a second fate of glass formers accompanied by intrinsic macroscopic inhomogenization. Therefore, instability towards crystallisation or phase separation imposes stringent limits on the ability to access the low-temperature equilibrium glassy states, preventing us from solving the Kauzmann paradox of glass transition[10,11].

Recently, there has been a remarkable breakthrough, i.e., the development of a novel model glass former with an extraordinary glass-forming ability that can avoid crystallisation and phase separation[22,28], allowing us to approach the vicinity of the hypothetical ideal glass transition. However, a disordered liquid state generally tends to lower its free energy through some orderings. Thus, the fundamental question remains whether other types of structural ordering may set in at low temperatures and what kind of roles they would play in the physics of supercooled liquids and glass transition.

In this article, we explore the above question by computer simulations of this model glass former[22,28]. Using real-space visualisation and quantitative structural analyses, we show that this system exhibits exotic compositional order at low temperatures, leading to the coexistence between a system-spanning network-like structure formed by the largest and smallest particles and patches formed by medium-size particles in its pores. Such exotic compositional order is not reflected in the conventional structural characterisations, e.g. the static structure factor that is useful in identifying crystallisation and phase separation. Furthermore, this ordering is not accompanied by any thermodynamic signature. Therefore, it has been unnoticed in previous studies. We find that the emergence of exotic compositional order directly impacts structural relaxation dynamics. Based on these findings, we discuss the roles of unconventional structural ordering in understanding glass transition.

## Results

We explore unconventional structural ordering at low temperatures of an extraordinarily good glass former, which is becoming a standard model for studying the glass transition problem[22,28]. Two key elements of this model glass former are the high particle-size polydispersity and the nonadditive interactions, which preclude the crystallisation and phase separation, respectively (see Methods for details). It thus provides a perfect platform to search for new types of structural ordering. The high particle-size polydispersity also dramatically enhances the efficiency of the swap Monte Carlo algorithm (SMC)[29], which allows the equilibration of the system at unprecedented low temperatures[22]. We generate equilibrium configurations using SMC over a broad range of temperatures, from the simple-liquid regime down to $T = 0.03$. This is below the hypothesised ideal glass transition temperature $T_0 = 0.067$ according to the Vogel–Fulcher–Tammann fitting of the structural relaxation time (see Supplementary Fig. 5), which is studied by standard molecular dynamics simulations.

### Exotic compositional order

We first visualise the typical structures of the system at high and low temperatures in Fig. 1. Without the assistance of further information, the bare particle configurations look disordered, and it is difficult to tell them apart [see Fig. 1a, d]. The structure factor $S(k)$ also shows no signs of ordering (see Supplementary Fig. 1). We then colour the particles according to their coordination numbers $z$ in Fig. 1b, e. While the high-temperature configuration appears random, the low-temperature one shows prominent features of complex compositional order. Particles with $z > 6$ and $z < 6$ correspond roughly to those with the largest and smallest sizes, respectively (for $T = 0.03$, the average diameters are 1.28, 0.95, 0.78 for particles with $z > 6, z = 6$, and $z < 6$, see Supplementary Figs. 2–4 for the further information). They tend to connect and form network-like structures. On the other hand, the network pores are filled with particles with $z = 6$, which correspond roughly to those with medium sizes. Since particles tend to have high $\Psi_6$ for $z = 6$ whereas low $\Psi_6$ for $z \neq 6$, we further colour the particles according to

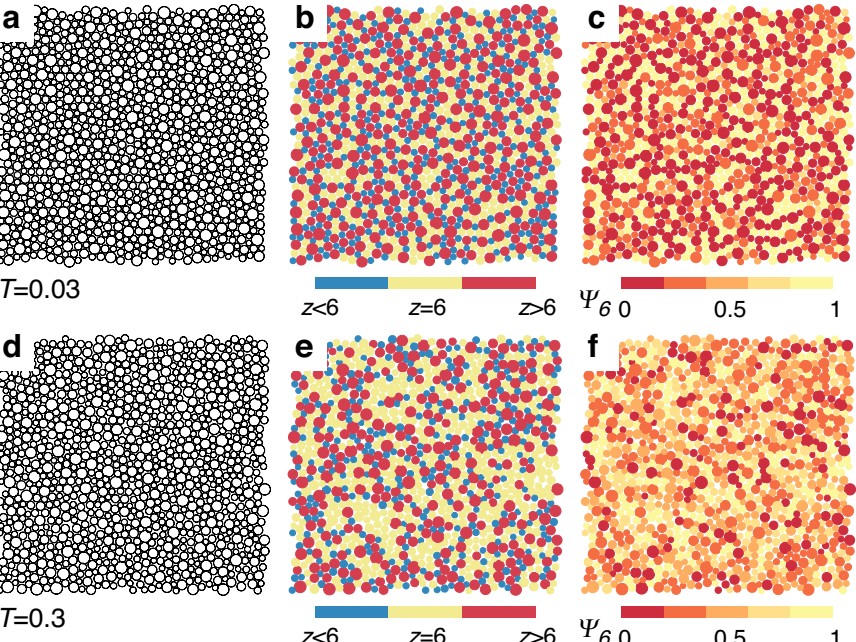

**Fig. 1 | Visualisation of the exotic compositional order.** Top panels: an equilibrium configuration at a low temperature ($T = 0.03$) is shown without colouring (**a**), coloured according to the coordination number $z$ (**b**), and according to the hexatic bond-orientational order $\Psi_6$ (**c**). In comparison, bottom panels (**d**–**f**) show the same plots as in (**a**–**c**) for an equilibrium configuration at a high temperature ($T = 0.3$). The exotic compositional order at low temperatures is invisible from the raw particle configurations but evident after proper colour coding.

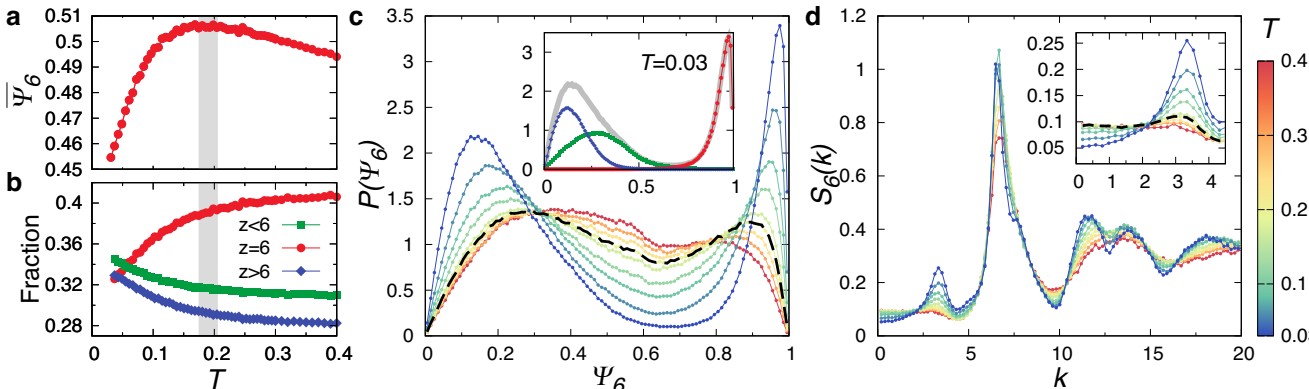

**Fig. 2 | Structural features of the exotic compositional order.** The temperature dependence of the average hexatic order $\overline{\Psi}_6$ (**a**) and the fraction of particles with different coordination numbers $z$ (**b**). The crossover where the exotic compositional ordering takes place is indicated by the grey bar at $T_\mu \approx 0.19$. **c** The probability distribution of hexatic order $P(\Psi_6)$ for a range of temperatures. Inset: $P(\Psi_6)$ at $T = 0.03$ (grey) is shown together with conditional probability distributions of $\Psi_6$ for particles with $z < 6$ (green), $z = 6$ (red), and $z > 6$ (blue). **d** The static structure factor according to the hexatic order, $S_6(k)$, for a range of temperatures. Inset: Magnified view on the emergence of a peak at $k \approx 3.1$. Panels **c**, **d** show results at $T = 0.4$, $0.3$, $0.24$, $0.19$, $0.17$, $0.12$, $0.09$, $0.06$, and $0.03$, with that at $T_\mu$ highlighted by dashed black curves (note that they share the same colour bar).

$\Psi_6$ in Fig. 1c and f. This colouring highlights the two phases formed at low temperatures, the network-like structures with low $\Psi_6$ and the conjugate patches of particles with high $\Psi_6$. Compositional order is often seen in simple mixtures or polydisperse systems as the tendency of particles with particular characters to stay together, leading to local structure ordering[27]. Oxide glasses like $SiO_2$ have network structures, but which results from covalent chemical bonding[30]. The observed network-like structure filled with patches in a soft repulsive system with continuous polydispersity is unusual and has not been revealed in previous studies. We, therefore, name this unconventional mesoscopic structural order *exotic compositional order*. It fundamentally differs from the ordinary phase separation featured by macroscopic inhomogeneity. Also, although sharing a similar morphological feature, it differs essentially from microphase separation often observed in block copolymers due to the lack of any distinct thermodynamic signature (we have checked the energy, pressure, and specific heat as a function of temperature)[31,32].

We then characterise the structural features upon the emergence of exotic compositional order. Figure 2a shows the temperature dependence of the average hexatic order parameter $\overline{\Psi}_6$, which has a peak at around $T_\mu \approx 0.19$, indicating a crossover of the underlying structural order. Correspondingly in Fig. 2b, below $T_\mu$, the fraction of particles with $z = 6$ ($z \neq 6$) shows a faster downturn (upturn) from a gentle evolution with decreasing temperature. This can be understood as a result of the competition between thermal fluctuations and structural ordering (see also Supplementary Fig. 2). The thermal fluctuation dominates for $T > T_\mu$; therefore, the hexatic order optimises gently with decreasing temperature. Whereas for $T < T_\mu$, the structural ordering takes control. More $z \neq 6$ particles appear during the formation of system-spanning networks, whereas $z = 6$ particles decrease and organise into small patches; both effects contribute to the decrease of $\overline{\Psi}_6$. Figure 2c shows the probability distribution of the hexatic order parameter, $P(\Psi_6)$, for a range of temperatures. The bimodal feature of $P(\Psi_6)$ is mild at high temperatures but develops significantly below $T_\mu$, which is a marked signature of structure ordering with coexistence between high-$\Psi_6$ droplet-like and low-$\Psi_6$ network-like structures. The inset of Fig. 2c shows $P(\Psi_6)$, together with the conditional probability distributions, for particles with different $z$ at $T = 0.03$. Clearly, particles with $z = 6$ and $z \neq 6$ contribute respectively to the high- and low-$\Psi_6$ peaks in $P(\Psi_6)$. Therefore, we can use the hexatic order parameter $\Psi_6$ to binarise the system and effectively uncover the exotic compositional order. We thus characterise the morphology of the exotic compositional order using the static

structure factor according to the hexatic order, $S_6(k)$ (see Methods for its definition). As shown in Fig. 2d, a peak emerges at $k \approx 3.1$ and grows with decreasing temperature, signifying the emergence of patches of high-$\Psi_6$ particles among the low-$\Psi_6$ network. The inset of Fig. 2d further magnifies the rapid growth of the peak below $T_\mu$. This, together with the direct visualisation shown in Fig. 1, provide strong pieces of evidence of the exotic compositional order in the system.

To further illustrate the exotic compositional order, we separately characterise the partial static structure factors for particles with $z = 6$ and $z \neq 6$. Figure 3a shows $S_{z=6}(k)$ over a range of temperatures. Similar to the static structure factor for the hexatic order, $S_6(k)$, $S_{z=6}(k)$ shows a peak at $k \approx 3.1$ below $T_\mu$, which grows with decreasing temperature. It indicates the emergence of patches of $z = 6$ particles. The similar behaviour between $S_6(k)$ and $S_{z=6}(k)$ is reasonable since there is a close correspondence between particles with $z = 6$ and high values of $\Psi_6$. Correspondingly, $S_{z\neq6}(k)$ shown in Fig. 3b characterises the emergence of network-like structures observed in Fig. 1b. We can see that $S_{z=6}(k)$ and $S_{z\neq6}(k)$ at $k \to 0$ decrease below $T_\mu$, suggesting that the large-scale fluctuations associated with the network-like structure are suppressed. This result, therefore, further confirms that the exotic compositional order is the coexistence state of the space-spanning network-like structures of $z \neq 6$ particles and the patches of $z = 6$ particles below $T_\mu$.

## Dynamic crossover

Using standard MD simulations, we further explore the influence of exotic compositional order on structural relaxation dynamics. Since particles with different sizes are observed to play different roles in the exotic compositional order, in addition to the global behaviour, we also characterise separately the structural relaxation of particles with the smallest, medium, and largest sizes to see how the exotic compositional order is coupled with particle mobility (see Methods for the details). Figure 4a shows the temperature dependence of the structural relaxation time $\tau_\alpha$ for the whole system and those for particles with different sizes. The high-temperature data can be fitted according to the Arrhenius law $\tau_\alpha = \tau_0 \exp(\Delta E / T)$. The onset of non-Arrhenius behaviour of $\tau_\alpha$ is found to coincide with the crossover temperature $T_\mu$ of the exotic compositional ordering, suggesting that the emergence of this unconventional structural ordering is correlated with a significant increase of the energy barrier for particle motion and a loss of fluidity below $T_\mu$. Another important and unexpected feature seen in Fig. 4a is that particles with quite different sizes, and correspondingly in different local structures, have very similar $\tau_\alpha$ even at low temperatures below $T_\mu$. To further exploit this point, the ratios of $\tau_\alpha$'s for

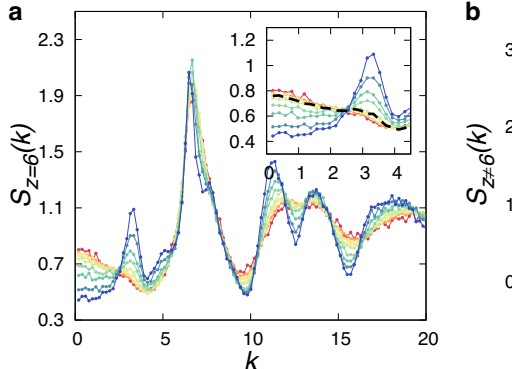
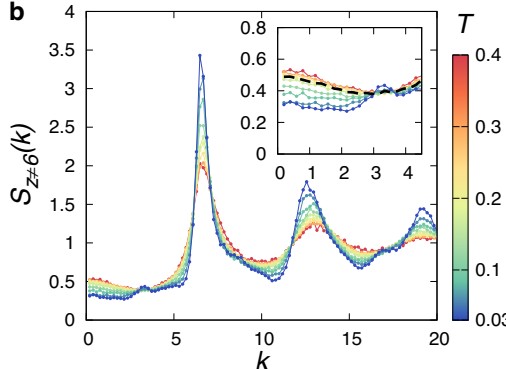

**Fig. 3 | Exotic compositional order revealed by partial static structure factor.** Partial static structure factors for particles with $z = 6$ (**a**) and $z \neq 6$ (**b**). Insets: Magnified view in the low-$k$ region. Dashed black curves highlight data at the crossover temperature of the exotic compositional ordering $T_\mu$. Results are shown for the same set of temperatures as Fig. 2d.

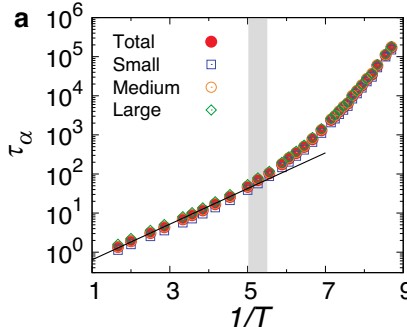
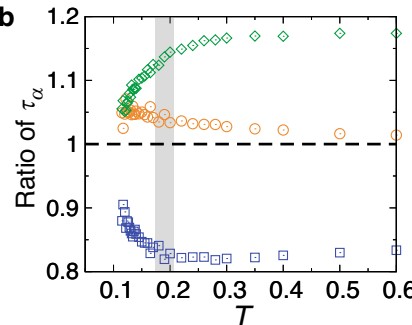

**Fig. 4 | Impact of the exotic compositional order on dynamics. a** Temperature dependence of the structural relaxation time $\tau_\alpha$ from standard MD simulations, for the whole system, and for particles with the smallest, medium, and largest sizes (32, 39 and 29%, respectively) (see Methods for the details). High-temperature data points of $\tau_\alpha$ for the whole system are fitted according to the Arrhenius law $\tau_\alpha = \tau_0 \exp(\Delta E / T)$ (solid line). **b** Temperature dependences of the ratios of $\tau_\alpha$'s for particles with the smallest, medium, and largest sizes with respect to $\tau_\alpha$ for the whole system. The dashed line corresponding to the ratio of 1 is a reference for eyes. The crossover temperature $T_\mu$ of the exotic compositional ordering is indicated by the grey bar in both panels.

particles with the smallest, medium, and largest sizes to $\tau_\alpha$ for the whole system are shown in Fig. 4b. The ratios are almost constant above $T_\mu$ and remain close to one for the whole temperature range under study. Very interestingly, the ratios for largest and smallest particles approach one below $T_\mu$, instead of deviating from one as usually observed in polydisperse colloidal glass formers[33,34]. It also differs from the corresponding 3D model system with the same interacting potential and particle-size distribution, for which the particle mobility strongly depends on the size[35]. We expect that such a peculiar structural relaxation behaviour originates from unconventional structural ordering. Below $T_\mu$, particles with different sizes are more involved in the correlated exotic compositional order and, therefore, necessary to relax cooperatively. This unusual dynamic feature is worthwhile being explored in more detail to understand the peculiarity of the current 2D model system.

## Thermodynamic aspect

Here, we characterise the thermodynamic aspect of the exotic compositional ordering and glass transition in our system by measuring the temperature dependence of the specific heat at constant volume, $C_v$. Two different methods were used to measure $C_v$. The first method is based on the standard definition of specific heat as the derivative of thermal energy per particle with respect to temperature, $C_{v,0} = (dE/dT)/N$. The second method is based on the fluctuation of thermal energy: $C_{v,f} = (\langle E^2 \rangle - \langle E \rangle^2)/Nk_B T^2$[36,37]. All characterisations are based on samples equilibrated using SMC. These two methods give consistent results in equilibrium simulations when the system is

ergodic. In our simulations, the ensemble average was carried out by time averaging in two ways: Monte Carlo (MC) simulations with and without the swap of particle diameters. A time scale is naturally involved in the simulations, corresponding to the physical time scale over which an actual measurement is performed in standard MC[38]. The results were then averaged over 100 independent realisations.

The temperature dependence of $C_v$ calculated by these two methods is shown in Fig. 5. We found that $C_{v,f}$ from SMC by a time average over $t = 10^7$ coincides with $C_{v,0}$ over the full range of temperatures under study, suggesting that SMC allows for sufficient canonical sampling of the whole phase space. However, $C_{v,f}$ obtained by standard MC shows a distinct deviation from $C_{v,0}$ at low temperatures. At shallow supercooling, the degree of deviation depends on the simulation time, converging at the lowest temperatures under study. Although characterised based on equilibrated samples, this behaviour is similar to that from the derivative of thermal energy by slow cooling using standard MD simulations (see the inset of Fig. 5). The approach of the results of standard MC to the Dulong-Petit law at low temperatures ($C_v = 2$ for solids in 2D) is a natural consequence of broken ergodicity due to the glass transition. On the other hand, the continuous increase in $C_v$ upon cooling for SMC indicates that SMC keeps equilibrating the system in a liquid state, at least down to the lowest temperature $T = 0.03$. Supposing that the VFT law is valid, these temperatures are already far below the hypothetical ideal glass transition temperature, $T_0 = 0.067$, where the physical structure relaxation time is expected to diverge. Another possible scenario is that the structural relaxation does not obey the VFT-like divergence but obeys the

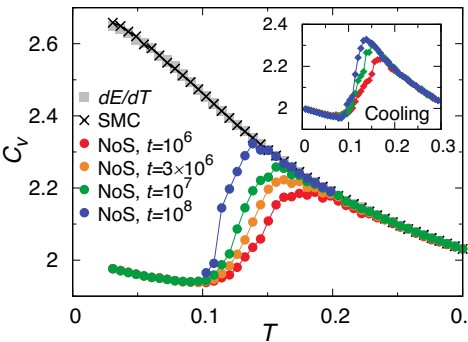

**Fig. 5 | Temperature dependence of specific heat $C_v$.** The specific heat capacity $C_v$ is measured by different protocols based on samples equilibrated using SMC. $C_v$ from the derivative of thermal energy per particle (from SMC) with respect to temperature is used as a reference ($dE/dT$). We also calculate $C_v$ based on the fluctuation-dissipation theorem, with the time average realised by SMC or standard MC algorithm (NoS indicating no swaps). Inset: For samples generated by slow cooling using standard MD simulations, $C_v$ is calculated from the derivative of thermal energy ($dE/dT$). From red to blue, the cooling rate is $\gamma = 6 \times 10^{-6}$, $6 \times 10^{-7}$ and $6 \times 10^{-8}$, respectively. In this case, $C_v$ from the fluctuation-dissipation theorem closely follows the derivation from thermal energy.

Arrhenius law at low temperatures[28]. Even in this case, the physical relaxation time at the lowest temperature goes far beyond the age of the universe[28]. Whether the exceptional stability of this system is related to the exotic compositional ordering is an intriguing question that is linked to whether we can consider the behaviour typical of ordinary glass-forming liquids.

These results also show no thermodynamic signature of the exotic compositional ordering around $T_\mu \approx 0.19$. Since $T_\mu$ is located in a temperature region where the system can be equilibrated even without SMC, the exotic compositional ordering is an intrinsic equilibrium property of this system and not due to SMC. In principle, such unconventional ordering can be induced by mixing more than two basins, including an ordered state whose free energies are degenerated, since SMC allows the system to overcome an extraordinarily high barrier. However, this may not be the case since the exotic compositional ordering already begins in a temperature region where the system can be equilibrated without SMC.

## Discussion

In this article, we explore the unconventional structural ordering at low temperatures of a novel model glass former designed to avoid crystallisation and phase separation, two well-known fates of most glass formers at low temperatures. Logically, the remaining possible structural ordering that one may expect is a certain kind of microscopic or mesoscopic ordering. This is indeed what we find in this study. It emerges as a coexistence of complex network-like structures formed by the largest and smallest particles ($z \neq 6$ and low $\Psi_6$) and patches formed by medium-size particles ($z = 6$ and high $\Psi_6$) surrounded by the network. Since this ordering does not involve density inhomogeneity, it has remained undercover by eye or conventional two-point correlation functions such as $S(k)$. An unusual dynamic crossover is found accompanying the emergence of this exotic compositional order. How should we understand such exotic compositional order in the context of the glass transition problem? Considering that there is no crystalline order or macroscopic inhomogeneity, is it acceptable as a special form of glassy order in exploring the generic nature of glass transition based on this system? Or, is it necessary to put the exotic compositional order on an equal footing with crystallisation and ordinary phase separation and, therefore, to be avoided in the study of glass physics? We hope that the answer to those fundamental questions raised from our findings may shed light on the basic understanding of glass transition.

## Methods
### Simulations
We simulate a model glass former with optimised particle-size polydispersity and interaction potential[22,28]. We focus on the two-dimensional systems for the ease of structure characterisations, which was recently studied in ref. 28. The particle diameter $\sigma$ follows the probability distribution $P(\sigma) = A\sigma^{-3}$ for $\sigma \in [0.73, 1.62]$ and zero otherwise. This corresponds to a large polydispersity $\delta = \sqrt{\langle\sigma^2\rangle - \langle\sigma\rangle^2}/\langle\sigma\rangle = 0.23$, which avoids the crystallization of the system. The interaction potential between particles $i$ and $j$ is given by $V(r_{ij}) = \epsilon(\sigma_{ij}/r_{ij})^{12} + f(r_{ij})$, when $r_{ij}/\sigma_{ij} < 1.25$ and zero otherwise. Here $f(r_{ij}) = c_0 + c_2(\sigma_{ij}/r_{ij})^2 + c_4(\sigma_{ij}/r_{ij})^4$ ensures continuity of both potential and force at the cutoff $r_{ij}^c = 1.25\sigma_{ij}$, and $\sigma_{ij} = (\sigma_i + \sigma_j)(1 - \Delta|\sigma_i - \sigma_j|)/2$. We set $\Delta = 0.2$, corresponding to a nonadditive mixing rule, which avoids the phase separation of large and small particles. All particles have the same mass $m$, and the number density is set $\rho = 1$. The length, energy, and temperature are in units of the averaged diameter $\langle\sigma\rangle$, $\epsilon$, and $\epsilon/k_B$, where $k_B$ is the Boltzmann constant. The systems are equilibrated using the swap Monte Carlo algorithm (SMC)[22]. Specifically, we prepare the configurations by slow cooling in a stepwise fashion from a high-temperature equilibrated state. The systems are equilibrated at the target temperature for at least 40 times the structural relaxation time with SMC[28], in addition to the extra equilibration during the slow cooling process. The obtained results are compared with those based on an ensemble using 1/10 of the equilibration time, the convergence of which confirms the equilibration. The time unit in Monte Carlo simulations (MC) is chosen as one full set of operations of all particles. Simulations are performed in square boxes with periodic boundary conditions. We mainly study systems with $N = 1000$ particles and confirm the absence of finite-size effects using $N = 10000$.

### Analysis of structure
We employ the radical Voronoi tessellation to characterise each particle's local environment, which properly takes into account the large polydispersity[39]. Neighbouring particles of a particle are identified as those sharing an edge of the Voronoi cells with the particle. Since we find that particles with $z = 6$ neighbours play a conjugate role to the others, we describe the local structure by the hexatic bond-orientational order parameter $\Psi_6$[40]. For particle $j$, we have $\Psi_6^j = |\sum_k e^{6i\theta_{jk}}/n_j|$, where $n_j$ is the number of nearest neighbours of particle $j$, and $\theta_{jk}$ is the angle of the bond $\mathbf{r}_{jk} = \mathbf{r}_k - \mathbf{r}_j$ with respect to the x-axis. We emphasise that here high $\Psi_6$ does not indicate good structural order but simply a sixfold rotational symmetry (roughly $z = 6$), whereas low $\Psi_6$ indicates a local environment close to other folds of rotational symmetry ($z \neq 6$). The global structure of the system is characterised using the static structure factor $S(k) = |\sum_i \exp(i\mathbf{k} \cdot \mathbf{r}_i)|^2/N$ (see Supplementary Fig. 1). Here, $\mathbf{k}$ is the wavevector. To characterise the spatial structure of $\Psi_6$ field, which sensitively detects the order of this system, we calculate the modified static structure factor according to the hexatic order $S_6(k) = |\sum_i \Psi_6^i \exp(i\mathbf{k} \cdot \mathbf{r}_i)|^2/N$, which is the Fourier transform of the corresponding two-point correlation function[27]. Moreover, the partial static structure factors are also characterised for particles with $z = 6$ and $z \neq 6$ due to their different roles in the exotic compositional order. The partial static structure factor for particles with $z = 6$ is defined as $S_{z=6}(k) = |\sum_i' \exp(i\mathbf{k} \cdot \mathbf{r}_i)|^2/N_6$, with $\sum'$ indicating the summation over only particles with $z = 6$ and $N_6$ being the number of those particles. The partial static structure factor for particles with $z \neq 6$ is defined accordingly.

### Analysis of dynamics
We study the physical evolution of the systems using molecular dynamics (MD) simulations. The time unit in molecular dynamics simulations is $\sqrt{m\langle\sigma\rangle^2/\epsilon}$. To remove the influence of long-wavelength Mermin–Wagner fluctuations, we characterise the dynamics by relative positions $\underline{\mathbf{r}}_j(t) = \mathbf{r}_j(t) - \sum_k \mathbf{r}_k(t)/n_j$, where the summation is over all

nearest neighbours of particle $j$[41–43]. The self-intermediate scattering function is then given as $F_s(k,t) = \langle \sum_j \exp(i\mathbf{k} \cdot [\underline{\mathbf{r}}_j(t) - \underline{\mathbf{r}}_j(0)]) / N \rangle$, with $k = |\mathbf{k}|$ being around the first peak of the static structure factor. The structural relaxation time $\tau_\alpha$ is defined from $F_s(k, \tau_\alpha) = 1/e$. It is possible to fit the temperature dependence of $\tau_\alpha$ according to the Vogel–Fulcher–Tammann (VFT) law with the VFT ideal glass transition temperature $T_0 = 0.067$ (see Supplementary Fig. 5). Motivated by the observation that particles with different sizes play different roles in the exotic compositional order (see Fig. 1), we also characterise the partial self-intermediate scattering function for particles with the smallest, medium, and largest sizes (32, 39 and 29%, respectively. See Supplementary Fig. 6.). Considering the close correspondence of particle-size and coordination number (see Supplementary Fig. 2b), these fractions are chosen according to Fig. 2b at $T_\mu$, and the results are found insensitive to slight variations of the values.

## Data availability
The data that support the findings of this study are available from the corresponding authors upon request.

## Code availability
The codes that are used to generate results in the paper are available from the corresponding authors upon request.

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

## Acknowledgements

We thank Misaki Ozawa and Trond S. Ingebrigtsen for discussions related to this work and Xinyu Fan for cross-checking the results of the partial self-intermediate scattering function. We also express our sincere gratitude to the reviewers for their invaluable and constructive feedback during the review process of this paper. This work was supported by the National Natural Science Foundation of China (Project 12274392, H. Tong) and by Specially Promoted Research (JP20H05619, H. Tanaka) and Scientific Research (A) (JP18H03675, H. Tanaka) from the Japan Society of the Promotion of Science (JSPS).

## Author contributions

H. Tong and H. Tanaka conceived the project, H. Tong performed research, and H. Tong and H. Tanaka discussed and wrote the manuscript.

## Competing interests

The authors declare no competing interests.
