## [Peer Review File · Nature Communications]

Emerging exotic compositional order on approaching low-temperature equilibrium glassesREVIEWER COMMENTS

Reviewer #1 (Remarks to the Author):

This is a very interesting paper that analyzes the structure and dynamics of a glass-former that has been recently introduced and studied extensively in the literature. This glass-former was introduced with the aim of having a model in which the SWAP algorithm is effective (ie one can get equilibrated configurations at low temperature) and that behaves as a realistic super-cooled liquid. Until now, it was thought that no structural ordering was taking place in this system. The present manuscript reveals that this is not the case. It shows that microphase separation takes place and its connection to dynamical properties.

Given the impact and the large number of works based on these liquids for which SWAP works, the present study is very important. It casts doubts on their applicability to study the glass transition problem, it gives guidelines for future studies, and possibly suggests the reason why SWAP works for these systems. I certainly recommend it for publication, but after that the following point is satisfactorily taken into account:

-I don't understand all the discussion about the specific heat. Even if the initial configuration is equilibrated the system needs to decorrelate along the dynamical trajectory to get a correct measure of the energy fluctuation. So I don't see why the results in Fig 4 are surprising and need to be discussed. Clearly, the trajectories are not long enough because the relaxation time is too large for the lowest temperatures. This behavior is expected on general grounds for MC without swap in the low temperature regime in which Swap works. I suggest to better motivate this part or take it out.

Minor: It would be interesting to discuss/speculate if the emergence of the microphase separation with a system spanning network is what makes SWAP work.

Reviewer #2 (Remarks to the Author):

This paper presents a computational study of a 2d polydisperse mixture that exhibits glassy behavior. The model can be equilibrated down to very low temperature using the swap Monte Carlo method and has been recently used to investigate some of the outstanding issues about the glass transition. The authors report that a subtle form of ordering develops in this mixture as the temperature is lowered. They argue about possible implications of their findings for the low-temperature fate of glass-forming liquids.

The results contained in Fig.1, 2 and 4 are indeed interesting and warrant publication in some form. They may trigger some further analysis of the structure of polydisperse glassy models, which could be of interest to the theoretical glass community. Unfortunately, the authors try too hard to make up a story about what they call "exotic microphase separation" and provide loose arguments for several of their claims.

My overall assessment is that this is an interesting study, which has however some presentation flaws that currently make it unsuitable for publication. I recommend the authors to address the points below and resubmit a toned-down manuscript.

Main issues:

1) The central claim of the paper is that "exotic microphase separation" may be a third possible low-temperature "fate" of glass-forming liquids, along with (i) an ideal glass transition or (ii) crystallization and/or phase separation. While this argument may sound intriguing at first, it actually finds little support in the paper. In the context of glass transition

studies, fates (i) and (ii) are advocated at low temperature, i.e. below T_g , to bypass the so-called entropy crisis. However, compositional order in the studied model sets in progressively, with a crossover temperature T_μ which is only slightly below the onset of glassy dynamics and well above T_g . This is certainly an interesting "feature" of the model, but not a way to circumvent the entropy crisis, so this has little to do with its low temperature "fate". Since the configurational entropy of the model keeps decreasing at low temperature (at least these are the results of Ref. 28) the possible fates remain (i) and (ii).

2) The wording "exotic microphase separation", used here to describe the tendency of large particles to be surrounded by small particles, is misleading. Microphase separation, as encountered in polymers and colloids, typically refers to a phase transition or some sharp segregation, but the equilibrium data for the specific heat in fig.4 show a smooth behavior across T_μ , without any peak or singularity, and the evolution of all the indicators in fig.2 are also smooth across T_μ . In the absence of any evidence from finite size effects (not studied here), the observed ordering cannot be attributed to some underlying transition. Thus, it is not appropriate to think of it as microphase separation. The authors seem to be aware of this when they write that the observed "microphase separation does not exhibit any distinct thermodynamic signature. Thus, it essentially differs from microphase separation often observed in block copolymers". Adding "exotic" does not fix the problem for me.

I think the authors should avoid calling the observed ordering "microphase separation". (It would be fine to mention some analogy with it, though). The system simply develops "compositional order", which is neither exotic nor a signature of a phase transition. Many other glass-formers do (oxides, metallic glasses) in somewhat similar ways.

3) It is not clear whether the observations for this polydisperse model are specific to 2d or carry out to 3d. This is a central issue if one wants to make general statements about glassy behavior. Compositional order in the additive 3d variant of this model has been studied in detail in Ref. 27 and found to evolve smoothly as a function of packing fraction. It would be useful to make direct contact with the analysis of Ref. 27, to see whether compositional order in the 2d model has anything special. (To be clear: I am not suggesting to simulate the 3d model, just to apply a similar analysis as Ref. 27)

4) The authors analyze the T-dependence of the structural relaxation times in two different ways. Fig.3a suggests two distinct Arrhenius regimes, with a crossover around T_μ . Fig.S4b shows instead that the VFT equations fits well the full range of temperatures and carries no signature of a crossover. The authors give no reason to favor one representation over the other and yet choose to put the one showing a crossover upfront. Straight lines can be fitted anywhere locally in an Arrhenius plot so the analysis in Fig.3a has little quantitative support. On the other hand, there is an obvious inconsistency between the VFT fit, which gives $T_0=0.067$, and the fact that the swap MC equilibrates down to $T\sim 0.026$. These aspects should be presented and discussed more carefully.

The analysis of the Debye-Waller factor in Fig.3b is also questionable. In particular, the authors report a finite Debye-Waller factor of about 0.83 at temperatures at which there is not even a well-defined plateau in the correlation functions. The fitting exercise in Fig. S4 is not well documented and the results likely depend on the choice of the time range to fit the stretched exponential. The presence of a sharp kink is thus dubious.

5) The discussion about broken ergodicity and the reference to the work by Palmer is largely irrelevant for the purpose of this work. The fact that the specific heat from MD has a maximum that shifts with cooling rate is a trivial consequence of the kinetic glass transition on the time scales of conventional simulations. The fact that fluctuation and derivative expressions for C_V provide consistent results for swap MC is a strong hint that these simulations are indeed performed at equilibrium down to $T=0.026$ (as also shown in Ref. 28).

The fact that the swap MC results do not approach the Dulong-Petit limit is consistent with the fact that the system remains an ergodic liquid down to that temperature (as also seen in Ref. 28).

None of the authors' findings support the statement that "unphysical effects on other thermodynamic quantities may arise due to the unphysical particle movements included in SWAP" so this sentence should be removed. Also, the sentence on p.9: "The approach to the DulongPetit law at low temperatures ($C_v = 2$ for solids in 2D) further confirms the physical relevance of our results and highlights the importance of adequately taking into account broken ergodicity when considering the glass transition problem" is unclear: I do not see how a trivial result (a classical solid at low T behaves harmonically) should bring any insight into the glass transition problem.

Minor points:

- p.3: "of this novel model glass former" -> the model is not novel, it was introduced in Ref. 28
- p.3: SWAP is not an acronym, the method should be spelled "swap MC"
- p.4: "hypothesised ideal glass transition temperature $T_0 = 0.067$ " -> it should be mentioned that this value is obtained in the Supplemental Information
- p.7 and caption of Fig.4: "the derivation of thermal energy" -> "derivative"

Reviewer #3 (Remarks to the Author):

See pdf.

Review for "Exotic microphase separation: A third possible fate of glass formers" by H. Tong and H. Tanaka

January 27, 2023

Summary of the work

The paper by Tong and Tanaka presents computer simulations of a particular two-dimensional poly-disperse glass-forming system which has been introduced recently (with the advent of the SWAP method) to achieve extremely low-temperature equilibrium configurations that are stable against demixing and crystallization. The authors study structural and dynamical aspects of this novel system. The main claim of the paper is that so-called microphase separation occurs at very low temperatures, a possibility which had not been considered nor observed thus far. In particular the study suggests that at very low temperatures the system develops a particular structure where droplets with increased six-fold symmetry are immersed in a network of particles made up of the smallest and largest particles. The study of the dynamical properties of the system reveals the existence of a dynamical crossover at temperature $T_\mu \approx 0.168$ during the formation of the 'spanning network', beyond which rigidity is greatly enhanced. The authors also provide a discussion on the averaging problem in ergodic & non-ergodic systems and show results on heat capacity measurements. The paper ends by suggesting that microphase separation might be a third, hitherto unnoticed, possible fate of glass formers that also connects to several other important questions regarding the fundamental nature of the glass transition.

I believe the discovery of microphase separation in the polydisperse mixture employed in SWAP is an important and new finding on a very timely topic which, if true, will certainly carry important implications for state-of-the-art computer simulations of supercooled liquids. Indeed, in view of the rising popularity of the SWAP method to address many unsolved questions surrounding the glass transition at unprecedentedly low temperatures, it would be extremely important to be aware of previously unnoticed structural changes that may obscure, and perhaps retrospectively cast some doubt on, the physical interpretation of these state-of-the-art computer simulation studies. As such, the scope and potential impact of the work would certainly appeal to the broad readership of Nature Communications. However, upon critical reading, I am not convinced that the authors have demonstrated sufficiently rigorously that their main conclusions are supported by their data. I thus cannot recommend publication in Nature Communications in the present form, and urge the authors to seriously consider the comments below to more strongly substantiate their claims.

Fundamental issues

- The authors show results for samples equilibrated to temperatures as low as $T = 0.03$, which is half(!) the glass transition temperature and more than 2 times as low as previous (state-of-the-art) studies have studied [Scalliet, Guiselin, Berthier (2022)]. Yet the authors show no proof that their configurations are fully equilibrated. The authors should outline their equilibration

procedure in the Methods section in more detail and I strongly encourage the authors to show the structural relaxation time or mean squared displacement as a function of temperature during the SWAP equilibration step in order to confirm that they reach properly equilibrated samples.

- In Fig. 3(a), the authors show that the relaxation time of the system with respect to temperature can be fitted with two separate Arrhenius laws, with a crossover happening at temperature T_μ . Yet in the S.I. the authors also claim that the curve is equally well described by a VFT law, where there is no crossover temperature. This indicates that the identified ‘crossover’ that happens to coincide with T_μ is spurious, and could have been found at very different temperatures by changing the fit ranges. Indeed, it can already be seen by eye that the fits of Fig. 3(a) may lend themselves for somewhat arbitrary choices of the fitting parameters. I am thus not convinced that their data indicate that T_μ corresponds to the onset of the increase of the activation energy as the authors claim.
- Figure 3(b) at first glance seems to clearly indicate a significant change at T_μ , but I have concerns about how these data were obtained. Fitting stretched exponentials to measured data is notoriously difficult. At high temperatures, the plateau is in fact non-existent (see Fig. S4) and one could fit it equally well at any desired height. This modifies the value of the Debye-Waller factor of Fig. 3(b), thus rendering any physical interpretation impossible and the observed non-analytic behaviour across T_μ potentially meaningless. Furthermore at low temperatures, the data for the plateau height is not shown in Fig. S4(a), which makes the fit itself highly questionable. As it stands, I believe that the Debye-Waller factors shown in Fig. 3(b) are currently unreproducible and hence any conclusions drawn from them are unsupported.

Thus, the authors should ensure, unambiguously, that the non-analytic behaviour reported in Fig. 3(b) is not merely an artefact of the fitting procedure. For example, does the Debye Waller factor found depend on the initial guess of the fit? Moreover, the values for the KWW exponent β are unreported. The authors should check that they are consistent with each other and with the time-temperature superposition principle. The plateau height can also be obtained by other and arguably better means than fitting a stretched exponential, e.g. the authors can simply isolate the section of the intermediate scattering function with near-zero slope and determine the (averaged) corresponding height within some error margin, and the relaxation time can be determined by setting $F_s(k, \tau_\alpha) = 1/e$ for instance, thus removing the need of a complicated 3-parameter fit.

- The authors claim that their “results suggest that structural ordering in the form of exotic microphase separation leads to a significant loss of fluidity and a steep increase in the solidity below T_μ ”. The circumstantial evidence they present that the two phenomena both occur at T_μ is, I believe, based on questionable fitting procedures as outlined above, and thereby unconvincing. Additionally, a causal relation between the two is not established in the present work. In order to substantiate this statement and thereby one of the main claims of the paper, the authors must address my concerns with the fitting procedure and show that the emergence of the microphase separation causes loss of fluidity. For instance, the authors should show partial intermediate scattering functions $F_{z=6}(k, t)$ and $F_{z \neq 6}(k, t)$. If what the authors claim is true, these should display interesting changes at T_μ , thereby explicitly showing the change in dynamics due to the found microphase separation.
- The relevance of the section on the broken ergodicity via measurements of the heat-capacity with regards to the first part of the manuscript is unclear in the present form. The authors

should explain the role of T_μ , which is indicated in Fig. 4 but not discussed in the context of microphase separation in neither the main text nor the S.I. Do the authors imply that T_μ is the point beyond which non-SWAP dynamics give the wrong result due to local sampling only as the figure seems to suggest? This would be wrong, as one could simulate for even shorter times with no swaps and have the curve fall off at $T > T_\mu$.

I suggest that the authors reformulate this section in a clearer and relevant way to the microphase separation they observe, or remove it entirely from the paper, as it pertains to a different topic: that of *averaging in near or non-ergodic systems*.

- At the very end of the results, the authors write “Our findings thus evidence that, apart from dynamics, unphysical effects on other thermodynamic quantities may arise due to the unphysical particle movements included in SWAP”. This is technically true in the case that a SWAP simulation were over a time-scale much longer than a typical experiment, thereby ergodically exploring a system that on experimental timescales is not ergodic. However, I believe the statement may be misleading, since the effective time-scales over which experiments are typically conducted are much closer than those effectively obtained through SWAP than through standard simulation methods. This means that the degree of ergodicity experienced in SWAP is practically much closer to the experiments than in standard simulations, which implies that not SWAP, but rather standard MD simulations should require a careful account for this ergodicity discrepancy.

Minor comments and suggestions

- The authors use as quantifier of structure $S_6(k)$, which is a correlation function between density fluctuations which are weighed by the hexatic bond-orientational order parameter Ψ_6 . This should be appropriately referenced. It would be instructive to include a more in depth physical interpretation of this quantity. In particular this would help the reader have a better understanding of Fig. 2(d) where a peak emerges at low k and why/how it signals microphase separation. Additionally it would also help in the interpretation of the double peak observed between $10 < k < 15$. For the lowest temperature $T = 0.03$, the structure factor (Fig. S1) appears to show a premise of the double peak in Fig. 2(d) as well as a modest peak at low wave-vectors. Overall the section could be made much more convincing by also referring to, and showing in the main text, explicitly the partial structure factors $S_{z=6}(k)$ and $S_{z\neq 6}(k)$, where a very clear picture of the reported exotic microphase separation is presented. Together with Fig. 2(c) the global picture of microphase separation would then be very convincing.
- The authors state that the microphase separation is between intermediately sized particles with 6-fold order and small/large particles with non-6-fold order. This relationship between the particle size and 6-fold order would be best illustrated by a figure that shows the relationship between D_i and Ψ_6^i .
- The authors claim that the microphase separation is between droplets of 6-fold order and a network of non-6-fold order. These droplets cause a small peak in the structure factor at $k \approx 3$, corresponding to lengthscales of about 2 particle diameters. This can be confirmed by Fig. 1b and 1c, where the yellow patches indeed have a characteristic size of around 2 particle sizes. Could the authors confirm that this is the right interpretation and if so, is 2 particle diameters large enough for a collection of particles to be called a droplet?

- The authors claim that there are no ‘distinct thermodynamic signatures’ of the microphase separation. Could the authors be more precise on what thermodynamic quantities they have checked?
- In the context of the microphase separation observed, have the authors studied the dynamical features of particles with coordination number $z = 6$ and the other particles with $z \neq 6$? For example, the authors could study a partial intermediate scattering function or a partial non-Gaussian parameter. I believe this could greatly strengthen the main claim of the paper, as it would directly correlate the key features of the newly discovered structure with its dynamics.
- In a recent paper [Scalliet, Guiselin, Berthier (2022)], the system studied in this work has been used to study a hypothesised facet of glassy dynamics known as ‘dynamic facilitation’. Could the authors perhaps comment on the relevance of their results with respect to this work vis-a-vis dynamic heterogeneity at very low temperatures?
- Finally, a minor comment, in many figures (2cd for instance), the authors display the legend in the form of a continuous color bar, while all the curves correspond to discrete values. The continuous color bar makes it very difficult to identify at which temperatures the simulations were performed exactly, thus hampering reproducibility. I suggest that they either label the different lines explicitly or put a list of studied temperatures in the Methods section.

The main revisions are summarised below:

- (1) Following the suggestions of Reviewers #2 and #3, we have performed new analyses of the dynamics, and the results further support our conclusion: New Fig. 4 in the revised main text.
- (2) Following the suggestion of Reviewer #3, the results of the partial static structural factor for particles with $z = 6$ and $z \neq 6$ are now included in the main text: New Fig. 3 in the revised main text.
- (3) Following the suggestion of Reviewer #2, the compositional order is analysed and discussed accordingly: New Fig. S4 in the revised Supplementary Information.
- (4) Following the suggestion of Reviewer #3, the relationship between particle size and the hexatic order is characterised: New Fig. S3 in the revised Supplementary Information.
- (5) The title of the manuscript is revised as “Exotic microphase ordering and broken ergodicity on approaching low-temperature equilibrium glasses”, to avoid the possible confusion with microphase separation often observed in block copolymers and highlight the key points of this work.
- (6) The section on broken ergodicity is largely revised to explain the underlying physics and the relevance of our characterisations in a clearer manner, according to the constructive comments of both Reviewers #2 and #3. We have also added a new Fig. S6 in the revised Supplementary Information to illustrate further the impact of the swap Monte Carlo algorithm on broken ergodicity.

The revised parts are highlighted in blue colour in the revised manuscript.

In the following, we respond point-by-point to all the reviewers' comments.

Response to Reviewer #1's Comments:

This is a very interesting paper that analyzes the structure and dynamics of a glass-former that has been recently introduced and studied extensively in the literature. This glass-former was introduced with the aim of having a model in which the SWAP algorithm is effective (ie one can get equilibrated configurations at low temperature) and that behaves as a realistic super-cooled liquid. Until now, it was thought that no structural ordering was taking place in this system. The present manuscript reveals that this is not the case. It shows that microphase separation takes place and its connection to dynamical properties.

Given the impact and the large number of works based on these liquids for which SWAP works, the present study is very important. It cast doubts on their applicability to study the glass transition problem, it gives guidelines for future studies, and possibly suggests the reason why SWAP works for these systems. I certainly recommend it for publication, but after that the following point is satisfactorily taken into account:

We thank the reviewer for such a positive assessment of our work. We also appreciate the valuable comments of the reviewer. Below please find our detailed response.

-I don't understand all the discussion about the specific heat. Even if the initial configuration is equilibrated the system needs to decorrelate along the dynamical trajectory to get a correct measure of the energy fluctuation. So I don't see why the results in Fig 4 are surprising and need to be discussed. Clearly, the trajectories are not long enough because the relaxation time is too large for the lowest temperatures. This behavior is expected on general grounds for MC without swap in the low temperature regime in which Swap works. I suggest to better motivate this part or take it out.

We thank the reviewer for this comment. The concept of broken ergodicity in the glass transition problem is indeed subtle. We explain that “the system needs to decorrelate along the dynamical trajectory to get a correct measure of the energy fluctuation” is a usual requirement for systems with a short intrinsic time scale, e.g., in the gas or normal liquids. However, when the system is effectively trapped in one basin of the whole phase space, we must carefully consider the effect of broken ergodicity. It might be illustrative to consider the Ising ferromagnet as an example. Below a critical temperature, the system may stay in one of the macroscopic states of positive or negative magnetisation [Such a spontaneous symmetry breaking is a typical case of broken ergodicity; see Palmer, Adv. Phys. 31, 669 (1982)]. In such a situation, if the canonical ensemble is employed to describe the system, i.e., allowing the magnetisation to flip in the infinitely long time limit, it is expected to predict a system with zero magnetisation on average. This is incorrect for the physical situation. Due to the diverging time scale at deep supercooling, broken ergodicity would also affect the theoretical description of the glass transition phenomenon. This has been explicitly discussed in the seminal paper on the random-first-order transition theory of glass transition by Kirkpatrick, Thirumalai, & Wolynes [Phys. Rev. A 40, 1045 (1989)], which shows the necessity of switching from the canonical ensemble to a restricted ensemble according to the broken ergodicity at low temperatures, for obtaining the system's 'physical' free energy and entropy. Our result of specific heat measurements is within these theoretical considerations.

In particular, the approach of C_v to the Dulong–Petit law ($C_v = 2$ for solids in 2D) firmly

confirms the physical relevance of our results. In contrast, the unphysical effects on thermodynamic quantities like the specific heat may arise due to the unphysical particle swaps included in SMC.

We have largely revised this part to discuss this point more clearly. We hope the reviewer will find our discussion convincing.

Minor: It would be interesting to discuss/speculate if it the emergence of the microphase separation with a system spanning network is what makes SWAP works.

We thank the reviewer for this valuable suggestion. We have added a discussion on this point at the end of the “Broken ergodicity” section. We speculate that the extremely high efficiency of SMC in this model might be related to the exotic microphase ordering with a continuous character. While structure relaxations are frozen out with physical dynamics in standard MD simulations, due to the continuous and extensive polydispersity, it is still possible to cross the substantial energy barriers by particle swaps, leading to the gradual evolution of the structure.

We hope that the reviewer will find that the concerns have been adequately addressed and that our revised manuscript has been significantly improved and now deserves publication in *Nature Communications*.

Response to Reviewer #2's Comments:

This paper presents a computational study of a 2d polydisperse mixture that exhibits glassy behavior. The model can be equilibrated down to very low temperature using the swap Monte Carlo method and has been recently used to investigate some of the outstanding issues about the glass transition. The authors report that a subtle form of ordering develops in this mixture as the temperature is lowered. They argue about possible implications of their findings for the low-temperature fate of glass-forming liquids.

The results contained in Fig. 1, 2 and 4 are indeed interesting and warrant publication in some form. They may trigger some further analysis of the structure of polydisperse glassy models, which could be of interest to the theoretical glass community. Unfortunately, the authors try too hard to make up a story about what they call "exotic microphase separation" and provide loose arguments for several of their claims.

My overall assessment is that this is an interesting study, which has however some presentation flaws that currently make it unsuitable for publication. I recommend the authors to address the points below and resubmit a toned-down manuscript.

We thank the reviewer for carefully reading our manuscript, appreciating our work, and providing valuable, constructive comments and criticisms. We also thank the reviewer for giving us an opportunity to revise the manuscript. Following the valuable suggestions, we have substantially revised our manuscript. We hope that the revised manuscript can better convey this work's messages. Below please find our detailed responses.

Main issues:

1. The central claim of the paper is that "exotic microphase separation" may be a third possible low-temperature "fate" of glass-forming liquids, along with (i) an ideal glass transition or (ii) crystallization and/or phase separation. While this argument may sound intriguing at first, it actually finds little support in the paper. In the context of glass transition studies, fates (i) and (ii) are advocated at low temperature, i.e. below T_g , to bypass the so-called entropy crisis. However, compositional order in the studied model sets in progressively, with a crossover temperature T_μ which is only slightly below the onset of glassy dynamics and well above T_g . This is certainly an interesting "feature" of the model, but not a way to circumvent the entropy crisis, so this has little to do with its low temperature "fate". Since the configurational entropy of the model keeps decreasing at low temperature (at least these are the results of Ref. 28) the possible fates remain (i) and (ii).

We thank the reviewer for this thoughtful comment. First, we explain that while the ideal glass transition is hypothesised from theoretical considerations, here we consider the low-temperature stable states accessible by physical processes which may intervene and impede the possible occurrence of ideal glass transition. Two well-known low-temperature fates of most glass-forming liquids are crystallisation (macroscopically homogeneous) and phase separation (macroscopically inhomogeneous) (or, their mixture). In this study, we explore the possibility of other types of structural ordering that may set in at low temperatures and interfere with the physics of supercooled liquids and glass transition. Logically, the remaining possible structural ordering that one may expect is a certain kind of microphase ordering.

Indeed, what we discover in current model glass former is a nonconventional structural ordering, which shares a similarity with microphase separation in morphology but differs from it in thermodynamic behaviors. We agree with the reviewer that it is not yet clear how to categorize this nonconventional structural ordering, and that is why we only discuss possible explanations of the observation and leave the question open for future studies. Now, following the reviewer's suggestions, we have revised the manuscript appropriately.

We also note that, as discussed explicitly in the seminal paper on the random-first-order transition theory of glass transition by Kirkpatrick, Thirumalai, & Wolynes [Phys. Rev. A 40, 1045 (1989)], it is crucial to switch from the canonical ensemble to a restricted ensemble according to the broken ergodicity at low temperatures to correctly obtain the system's 'physical' free energy and entropy. This point has been more clearly discussed in the revised manuscript. Therefore, we have reservations about the result of configurational entropy shown in Ref. 28.

2. The wording "exotic microphase separation", used here to describe the tendency of large particles to be surrounded by small particles, is misleading. Microphase separation, as encountered in polymers and colloids, typically refers to a phase transition or some sharp segregation, but the equilibrium data for the specific heat in fig.4 show a smooth behavior across T_{μ} , without any peak or singularity, and the evolution of all the indicators in fig.2 are also smooth across T_{μ} . In the absence of any evidence from finite size effects (not studied here), the observed ordering cannot be attributed to some underlying transition. Thus, it is not appropriate to think of it as microphase separation. The authors seem to be aware of this when they write that the observed "microphase separation does not exhibit any distinct thermodynamic signature. Thus, it essentially differs from microphase separation often observed in block copolymers". Adding "exotic" does not fix the problem for me.

I think the authors should avoid calling the observed ordering "microphase separation". (It would be fine to mention some analogy with it, though). The system simply develops "compositional order", which is neither exotic nor a signature of a phase transition. Many other glass-formers do (oxides, metallic glasses) in somewhat similar ways.

We thank the reviewer for this comment. We fully agree with the reviewer that the observed ordering may not be attributed to some underlying phase transition. We did not make such claims in our manuscript but admit that the wording was misleading. Following the reviewer's suggestion, we have revised the wording as "exotic microphase ordering" and stated it clearly as a crossover to avoid possible confusion with microphase separation often observed in block copolymers.

We note that, as shown in the following Figs. R1 & R2, the structural ordering observed in the current model appears different from the simple compositional order often observed in polydisperse hard spheres with additive interactions or other simple glass formers. Therefore, it may represent a special type of structural ordering due to the peculiarity of the current model glass former. This is the reason why we choose

the new wording.

3. It is not clear whether the observations for this polydisperse model are specific to 2d or carry out to 3d. This is a central issue if one wants to make general statements about glassy behavior. Compositional order in the additive 3d variant of this model has been studied in detail in Ref. 27 and found to evolve smoothly as a function of packing fraction. It would be useful to make direct contact with the analysis of Ref. 27, to see whether compositional order in the 2d model has anything special. (To be clear: I am not suggesting to simulate the 3d model, just to apply a similar analysis as Ref. 27)

We thank the reviewer for this valuable comment. Following the reviewer's suggestion, we have applied a similar analysis as Ref. 27 in the current model to characterise the local structure from the perspective of compositional order. Figure R1 shows the static structure factor of the diameter field $S_\sigma(k)$ and the diameter fluctuations field $S_{\delta\sigma}(k)$. Both $S_\sigma(k)$ and $S_{\delta\sigma}(k)$ show mild changes as a function of temperature, although $S_{\delta\sigma}(k)$ shows more complex features. Therefore, same as Ref. 27, the static structure factors of the diameter and diameter fluctuations fields are not sensitive to the compositional order.

Figure R2 shows the conditional probability distribution $P(\tilde{\sigma}|\sigma) = P(\tilde{\sigma}, \sigma) / P(\sigma)$ for a range of temperatures, which reveals interesting features of compositional order of the current model glass former. First, even at $T = 0.3$, which is much higher than the onset temperature of the system, there is already a clear negative correlation between σ and $\tilde{\sigma}$. This indicates a significant compositional order, i.e., a strong preference between particles with very different sizes even at high temperatures, which is expected to be the result of the nonadditive interaction. At the crossover temperature of the exotic microphase separation, additional features emerge at the low- σ side, becoming more significant at even lower temperatures. At $T = 0.03$, three sections can be clearly observed in $P(\tilde{\sigma}|\sigma)$, reflecting the three major components of the system, i.e., the small, medium, and large particles with $z < 6$, $z = 6$, and $z > 6$. This observation differs from Ref. 27 on a model of polydisperse hard spheres with additive interactions. Based on other characterisations in the manuscript, we expect small and medium-sized particles to have particular local structures. Therefore, the size ratios of the central to neighbouring particles are mostly constant, leading to the positive correlation in $P(\tilde{\sigma}|\sigma)$. For the largest particles, because of the large number of neighbours and the more irregular local structures, the competition between local ordering and the effect of nonadditivity leads to a slightly winding $P(\tilde{\sigma}|\sigma)$. Therefore, this is not a simple compositional order that particles may tend to be surrounded by neighbours with particular characters. Instead, it reflects that the

exotic microphase ordering is a peculiar feature of the model glass former.

We have included Fig. R2 as Fig. S4 and discussions accordingly in the revised Supplementary Information.

Fig. R1: Static structure factor of the diameter field $S_\sigma(k)$ (a) and the diameter fluctuations field $S_{\delta\sigma}(k)$ (b) over a range of temperatures. Dashed black curves highlight data at the crossover temperature of the exotic microphase ordering T_μ .

Fig. R2: Conditional probability distribution $P(\tilde{\sigma}|\sigma) = P(\tilde{\sigma}, \sigma) / P(\sigma)$ for a range of temperatures, as indicated by the labels.

4. The authors analyze the T-dependence of the structural relaxation times in two different ways. Fig.3a suggests two distinct Arrhenius regimes, with a crossover around T_μ . Fig.S4b shows instead that the VFT equations fits well the full range of temperatures and carries no signature of a crossover. The authors give no reason to favor one representation over the other and yet choose to put the one showing a crossover upfront. Straight lines can be fitted anywhere locally in an Arrhenius plot so the analysis in Fig.3a has little quantitative support. On the other hand, there is an obvious inconsistency between the VFT fit, which gives $T_0=0.067$, and the fact that the swap MC equilibrates down to $T\sim 0.026$. These aspects should be presented and discussed more carefully.

We thank the reviewer for this comment. We explain that the Arrhenius fitting of low-temperature structural relaxation times τ_α was motivated by Ref. 28. We agree with the reviewer that this is not a standard operation. According to the reviewer's comment and following the standard analysis process, in the revised manuscript, we use only the Arrhenius fitting of high-temperature τ_α and estimate the onset temperature by the deviation from the Arrhenius behaviour. We find that the onset temperature of the non-Arrhenius temperature dependence of τ_α is within the crossover temperature of the exotic microphase ordering. In some sense, this is not a surprising result because even in normal glass-forming liquids, the onset of non-Arrhenius behaviour is expected to be related to the development of a certain kind of structural order.

Concerning the inconsistency between the VFT fit, which gives $T_0=0.067$, and the fact that the swap MC equilibrates down to $T\sim 0.026$, we point out that this result is consistent with Ref. 28, and the inconsistency is only superficial. On the one hand, the VFT fit estimates the temperature T_0 at which the structural relaxation time is expected to diverge, with the extrapolation based on a particular theoretical scenario (e.g., the Adam-Gibbs theory). Similar to T_{MCT} by the fitting according to the mode-coupling theory, it provides a reference temperature from a theoretical perspective. Still, it does not ensure that different mechanisms may not set in and take control. In the current model glass former, due to the nonconventional structural ordering, it is not clear whether the VFT fitting would hold down to T_0 . On the other hand, the efficiency of swap MC originates from its nonphysical particle swaps, which do not respect the physical constraint of high energy barriers. In other words, it does not care whether the system is physically frozen (with realistic dynamics) within a particular basin of the phase space but instead can keep relaxing the structure by particle swaps. To better understand this point, we may consider a double potential well separated by an infinite energy barrier, with a particle in one of the wells with higher energy at the bottom. Physically, ergodicity is broken for the system, and the particle will stay in this well forever. However, if swap MC (or another nonlocal algorithm that does not respect the energy barriers) is employed, the particle may jump into the other well and get "relaxed". In this case, the algorithm is efficient (in finding the low energy states and visiting the whole phase space) on one side but unphysical on the other.

Fig. R3: A sketch of a double potential well separated by an infinite energy barrier.

The analysis of the Debye-Waller factor in Fig.3b is also questionable. In particular, the authors report a finite Debye-Waller factor of about 0.83 at temperatures at which there is not even a well-defined plateau in the correlation functions. The fitting exercise in Fig. S4 is not well documented and the results likely depend on the choice of the time range to fit the stretched exponential. The presence of a sharp kink is thus dubious.

We thank the reviewer for this very thoughtful comment. We performed the fitting following Ref. 43 of the previous manuscript without carefully checking the dependence of results on the fitting procedure. Thanks to this comment, we have performed the fitting more systematically. We find that the value of the Debye-Waller factor can be altered depending on the choice of the time range. To remove the flexibility in the choice of the time range, we have recalculated the self-intermediate scattering function further covering the short-time range, and tried to fit the full curve with a double KWW function: $F_s(k, t) = (1 - D) \exp\left(-\left(\frac{t}{\tau_1}\right)^{\beta_1}\right) + D \exp\left(-\left(\frac{t}{\tau_\alpha}\right)^{\beta_2}\right)$. While this gives a good fitting in ordinary glass-forming liquids [e.g., Simmons and Douglas, *Soft Matter* 7, 11010 (2011); Luo, et al., *PRL* 118, 225901 (2017)], we find that it fails to give a good fit of $F_s(k, t)$ in this model [see Fig. R4 below]. This may be related to the nonconventional structural ordering of this model. We further find that achieving a good fitting with a three KWW function is possible, but the physical picture behind this fitting is unclear. Therefore, to be rigorous, we have abandoned the results of the Debye-Waller factor in the revised manuscript. We greatly thank the reviewer for pointing out this point which helps us to correct this flaw in our previous analysis.

Following the suggestions of Reviewer #3, we have characterised the partial self-intermediate scattering function of the three major components emerging in the exotic microphase ordering, i.e., the smallest, medium, and largest particles (which correspond approximately to particles with $z < 6$, $z = 6$, and $z > 6$). Contrary to the simple expectation that τ_α of different components deviate more significantly from the average upon cooling, the approach to the average is observed below the crossover temperature of the exotic microphase ordering. This observation strongly indicates that, with the development of the exotic microphase ordering, particles with different sizes are more involved in such correlated structures and, therefore, necessary to relax cooperatively. This result is included in the new Fig. 4 of the revised manuscript,

which supports our conclusion that the development of the exotic microphase ordering is coupled with the dynamic crossover.

5. The discussion about broken ergodicity and the reference to the work by Palmer is largely irrelevant for the purpose of this work. The fact that the specific heat from MD has a maximum that shifts with cooling rate is a trivial consequence of the kinetic glass transition on the time scales of conventional simulations. The fact that fluctuation and derivative expressions for C_V provide consistent results for swap MC is a strong hint that these simulations are indeed performed at equilibrium down to $T=0.026$ (as also shown in Ref. 28). The fact that the swap MC results do not approach the Dulong-Petit limit is consistent with the fact that the system remains an ergodic liquid down to that temperature (as also seen in Ref. 28).

None of the authors' findings support the statement that "unphysical effects on other thermodynamic quantities may arise due to the unphysical particle movements included in SWAP" so this sentence should be removed. Also, the sentence on p.9: "The approach to the Dulong-Petit law at low temperatures ($C_V = 2$ for solids in 2D) further confirms the physical relevance of our results and highlights the importance of adequately taking into account broken ergodicity when considering the glass transition problem" is unclear: I do not see how a trivial result (a classical solid at low T behaves harmonically) should bring any insight into the glass transition problem.

We thank the reviewer for this thoughtful comment. The concept of broken ergodicity in the glass transition problem is indeed subtle, and the previous version may have failed to convey our message clearly. We fully agree with the review that "the specific heat from MD with a maximum that shifts with cooling rate is a trivial consequence of the kinetic glass transition on the time scales of conventional simulations", and we also agree that the swap MC allows a sufficient exploration of the whole phase space (regardless of the high energy barriers) and therefore gives C_V consistent with that from the derivative expressions. Consequently, with the dynamic rules of SMC, the system can be considered liquid even at the lowest temperatures in this study.

However, what we are concerned with in this section of the manuscript is the role of broken ergodicity in the **physically relevant** description of the low-temperature glassy state with realistic dynamics. In this case, the significant energy barriers may provide constraints to the realistic glassy dynamics (in addition to the protection of metastability of supercooled liquids), which is at the core of the glass transition problem. In essence, it would be a different problem if a system can cross any energy barriers and evolves with a certain kind of nonlocal dynamics.

At the lowest temperatures under study, the physical structure relaxation time is expected to diverge (below T_0 from the VFT law) or estimated to be far beyond the age of the universe (see Ref. 28, using an Arrhenius fitting of low-temperature τ_α). In such a situation, it is crucial to consider the broken ergodicity with special care to obtain physically relevant descriptions of the system. This point has been explicitly discussed in the seminal paper on the random-first-order transition theory

of glass transition by Kirkpatrick, Thirumalai, & Wolynes [Phys. Rev. A 40, 1045 (1989)]. It was shown necessary to switch from the canonical ensemble (average over the whole phase space) to a restricted ensemble (first average within each glassy basin and then average over different basins) according to the broken ergodicity at low temperatures, to obtain the system's 'physical' free energy and entropy. This further leads to the **physically relevant** theoretical description of the glass transition problem.

In this work, we study the effect of broken ergodicity through specific heat measurement. Focus is put on the lowest temperatures under study (with a diverging or cosmic timescale), which may be considered solid in any practical sense. Also, the ergodicity should be considered broken with any realistic dynamics. We find that C_v approaches the Dulong–Petit law using standard MC whereas that using SMC does not, suggesting the importance of properly considering the broken ergodicity to achieve a physically relevant description of the glassy system with realistic dynamics. This result, although appears trivial at first sight, actually exemplifies the impact of dynamic rules (realistic dynamics vs. particle swaps in SMC) on broken ergodicity and, therefore, the thermodynamic description of the system at low temperatures.

We have revised the manuscript substantially to state these points more clearly.

Minor points:

- p.3: "of this novel model glass former" -> the model is not novel, it was introduced in Ref. 28

We thank the reviewer for this comment. We note that, as stated explicitly in the above paragraph "Recently, there has been a remarkable breakthrough, i.e., the development of a novel model glass former...[22, 28]", we have no intention to claim that it is a novel model developed in this work. Now, we have removed the word "novel" to avoid possible misunderstandings.

- p.3: SWAP is not an acronym, the method should be spelled "swap MC"

We thank the reviewer for this comment. We have adopted the usage of "SWAP" from Ref. 28, which, as pointed out by the reviewer, may not be a proper acronym. Following the reviewer's suggestion, we have revised it as "SMC" as an acronym for "swap MC".

- p.4: "hypothesised ideal glass transition temperature $T_0 = 0.067$ " -> it should be mentioned that this value is obtained in the Supplemental Information

We thank the reviewer for this comment. We have now mentioned this point in the revised manuscript.

- p.7 and caption of Fig.4: "the derivation of thermal energy" -> "derivative"

We thank the reviewer for pointing out this typo, which has been corrected in the

revised manuscript.

We hope that the reviewer will find that all the concerns have been adequately addressed with the above explanations and that our revised manuscript has been significantly improved and now deserves publication in *Nature Communications*.

Response to Reviewer #3's Comments:

Summary of the work

The paper by Tong and Tanaka presents computer simulations of a particular two-dimensional poly-disperse glass-forming system which has been introduced recently (with the advent of the SWAP method) to achieve extremely low-temperature equilibrium configurations that are stable against demixing and crystallization. The authors study structural and dynamical aspects of this novel system. The main claim of the paper is that so-called microphase separation occurs at very low temperatures, a possibility which had not been considered nor observed thus far. In particular, the study suggests that at very low temperatures the system develops a particular structure where droplets with increased six-fold symmetry are immersed in a network of particles made up of the smallest and largest particles. The study of the dynamical properties of the system reveals the existence of a dynamical crossover at temperature $T_{\mu} \approx 0.168$ during the formation of the 'spanning network', beyond which rigidity is greatly enhanced. The authors also provide a discussion on the averaging problem in ergodic & non-ergodic systems and show results on heat capacity measurements. The paper ends by suggesting that microphase separation might be a third, hitherto unnoticed, possible fate of glass formers that also connects to several other important questions regarding the fundamental nature of the glass transition.

I believe the discovery of microphase separation in the polydisperse mixture employed in SWAP is an important and new finding on a very timely topic which, if true, will certainly carry important implications for state-of-the-art computer simulations of supercooled liquids. Indeed, in view of the rising popularity of the SWAP method to address many unsolved questions surrounding the glass transition at unprecedentedly low temperatures, it would be extremely important to be aware of previously unnoticed structural changes that may obscure, and perhaps retrospectively cast some doubt on, the physical interpretation of these state-of-the-art computer simulation studies. As such, the scope and potential impact of the work would certainly appeal to the broad readership of Nature Communications. However, upon critical reading, I am not convinced that the authors have demonstrated sufficiently rigorously that their main conclusions are supported by their data. I thus cannot recommend publication in Nature Communications in the present form, and urge the authors to seriously consider the comments below to more strongly substantiate their claims.

We thank the reviewer for reading our manuscript very carefully and for appreciating our work. We also thank the reviewer for the valuable, constructive comments, following which we have performed new analyses and substantially revised our manuscript. We are pleased that the new results further support our conclusions, and the discussions in the revised manuscript are now greatly strengthened. Below please find our detailed responses.

Fundamental issues

- The authors show results for samples equilibrated to temperatures as low as $T = 0.03$, which is half(!) the glass transition temperature and more than 2 times as low as previous (state-of-the-art) studies have studied [Scalliet, Guiselin, Berthier (2022)]. Yet the authors show no proof that their configurations are fully equilibrated. The authors should outline their equilibration procedure in the Methods section in more detail and I strongly encourage the authors to show the structural relaxation time or mean squared displacement as a function of temperature during the SWAP equilibration step in order to confirm that they reach properly equilibrated samples.

We thank the reviewer for this comment. We have employed exactly the same model glass former as Ref. 28 [Berthier et al., Nat. Commun. 10, 1 (2019)], where equilibration is realised down to $T=0.027$. The glass transition temperature estimated by VFT fitting of τ_α from standard MD simulations is also consistent with that from standard MC simulations in Ref. 28. Therefore, we have not achieved incredible equilibration beyond previous studies, which is also not the purpose of this work.

The equilibration can be realised with Ref. 28 as a reference. We have carefully set up the simulations accordingly and further confirmed the equilibration by checking the dependence of the results on the equilibration time. Specifically, we prepare the configurations at a target temperature by slow cooling in a stepwise fashion from a high-temperature equilibrated state. The structure relaxation time with SWAP at $T \approx 0.03$ is around $\tau_{\alpha, \text{swap}} \approx 2.5 * 10^6$ (Fig. S1 in the Supplementary Information of Ref. 28), and we have equilibrated the system for $t = 10^8 \approx 40\tau_{\alpha, \text{swap}}$ at $T=0.03$ in addition to the equilibration during the slow cooling process. The equilibration at higher temperatures is more sufficient considering the ratio of the equilibration time and the corresponding $\tau_{\alpha, \text{swap}}$. The results in the manuscript are compared with those based on an ensemble using 1/10 of the equilibration time. The convergence of results suggests efficient equilibration of the samples. We have now described the equilibration procedure in detail in the Methods section of the revised manuscript.

- In Fig. 3(a), the authors show that the relaxation time of the system with respect to temperature can be fitted with two separate Arrhenius laws, with a crossover happening at temperature T_μ . Yet in the S.I. the authors also claim that the curve is equally well described by a VFT law, where there is no crossover temperature. This indicates that the identified 'crossover' that happens to coincide with T_μ is spurious, and could have been found at very different temperatures by changing the fit ranges. Indeed, it can already be seen by eye that the fits of Fig. 3(a) may lend themselves for somewhat arbitrary choices of the fitting parameters. I am thus not convinced that their data indicate that T_μ corresponds to the onset of the increase of the activation energy as the authors claim.

We thank the reviewer for this valuable comment. The Arrhenius fitting of low-temperature structural relaxation times τ_α was motivated by Ref. 28, and we agree with the reviewer that this is not a standard operation. According to the reviewer's comment and following the standard analysis procedure, in the revised manuscript, we use only the Arrhenius fitting of high-temperature τ_α and estimate the onset temperature by the deviation from the Arrhenius behaviour. We find that the onset temperature of the non-Arrhenius temperature dependence of τ_α is within the crossover temperature of the exotic microphase ordering (since the exotic microphase ordering emerges gradually, there is a crossover covering a small range of temperatures). In some sense, this is not a surprising result because even in ordinary glass-forming liquids, the onset of non-Arrhenius behaviour is expected to

be related to the development of a certain kind of structural order.

- Figure 3(b) at first glance seems to clearly indicate a significant change at T_μ , but I have concerns about how these data were obtained. Fitting stretched exponentials to measured data is notoriously difficult. At high temperatures, the plateau is in fact non-existent (see Fig. S4) and one could fit it equally well at any desired height. This modifies the value of the Debye-Waller factor of Fig. 3(b), thus rendering any physical interpretation impossible and the observed non-analytic behaviour across T_μ potentially meaningless. Furthermore, at low temperatures, the data for the plateau height is not shown in Fig. S4(a), which makes the fit itself highly questionable. As it stands, I believe that the Debye-Waller factors shown in Fig. 3(b) are currently unreproducible and hence any conclusions drawn from them are unsupported.

Thus, the authors should ensure, unambiguously, that the non-analytic behaviour reported in Fig. 3(b) is not merely an artefact of the fitting procedure. For example, does the Debye-Waller factor found depend on the initial guess of the fit? Moreover, the values for the KWW exponent β are unreported. The authors should check that they are consistent with each other and with the time-temperature superposition principle. The plateau height can also be obtained by other and arguably better means than fitting a stretched exponential, e.g. the authors can simply isolate the section of the intermediate scattering function with near-zero slope and determine the (averaged) corresponding height within some error margin, and the relaxation time can be determined by setting $F_s(k, \tau_\alpha) = 1/e$ for instance, thus removing the need of a complicated 3-parameter fit.

We thank the reviewer for this comment. We agree with the reviewer that fitting stretched exponentials to the data is not an easy task. We performed the fitting following Ref. 43 of the previous manuscript without carefully checking the dependence of results on the fitting procedure. Thanks to this critical comment, we have performed the fitting more systematically. We find that the value of the Debye-Waller factor can be altered depending on the choice of the time range. To remove the ambiguity in the choice of the time range, we have recalculated the self-intermediate scattering function further covering the short-time range (previously, we have calculated $F_s(k, t)$ on the fly of the simulation, focusing on its behaviour over four time decades covering the structural (α -) relaxation), and tried to fit the full curve with a double KWW function: $F_s(k, t) = (1 - D) \exp\left(-\left(\frac{t}{\tau_1}\right)^{\beta_1}\right) + D \exp\left(-\left(\frac{t}{\tau_\alpha}\right)^{\beta_2}\right)$. While this gives a good fitting in ordinary glass forming liquids [e.g., Simmons and Douglas, *Soft Matter* 7, 11010 (2011); Luo, et al., *PRL* 118, 225901 (2017)], we find that it fails to give a good fit of $F_s(k, t)$ in this model [see Fig. R4 below]. Using τ_α defined by $F_s(k, \tau_\alpha) = 1/e$ as input does not improve the quality of fitting. This may be related to the nonconventional structural ordering of this model. We further find that achieving a good fitting with three KWW functions is possible, but the physical picture behind this fitting is unclear. As for the method of “isolate the section of the intermediate scattering function with near-zero slope”, we find it hard to determine the section with the near-zero slope with a consistent criterion, especially for medium and high temperatures.

Therefore, to be rigorous, we have abandoned the results of the Debye-Waller factor in the revised manuscript. We greatly thank the reviewer for pointing out this point which helps us to correct this flaw in our analysis.

Fig. R4: The self-intermediate scattering function at $T=0.15$ is fitted with a double KWW function $F_s(k, t) = (1 - D) \exp\left(-\left(\frac{t}{\tau_1}\right)^{\beta_1}\right) + D \exp\left(-\left(\frac{t}{\tau_\alpha}\right)^{\beta_2}\right)$ (a) and a three KWW function $F_s(k, t) = (1 - D - B) \exp\left(-\left(\frac{t}{\tau_1}\right)^{\beta_1}\right) + B \exp\left(-\left(\frac{t}{\tau_2}\right)^{\beta_2}\right) + D \exp\left(-\left(\frac{t}{\tau_\alpha}\right)^{\beta_3}\right)$ (b). Obvious systematic deviations of the fitting to the data points can be seen for the double KWW fitting, whereas the three KWW fitting describes the data points reasonably well.

- The authors claim that their “results suggest that structural ordering in the form of exotic microphase separation leads to a significant loss of fluidity and a steep increase in the solidity below T_μ ”. The circumstantial evidence they present that the two phenomena both occur at T_μ is, I believe, based on questionable fitting procedures as outlined above, and thereby unconvincing. Additionally, a causal relation between the two is not established in the present work. In order to substantiate this statement and thereby one of the main claims of the paper, the authors must address my concerns with the fitting procedure and show that the emergence of the microphase separation causes loss of fluidity. For instance, the authors should show partial intermediate scattering functions $F_{z=6}(k, t)$ and $F_{z \neq 6}(k, t)$. If what the authors claim is true, these should display interesting changes at T_μ , thereby explicitly showing the change in dynamics due to the found microphase separation.

We thank the reviewer for this very thoughtful comment. As explained in the responses to the above two comments, we choose not to fit stretched exponentials to the data but only determine τ_α from $F_s(k, \tau_\alpha) = 1/e$. Following a standard procedure, now we only fit the high-temperature τ_α according to the Arrhenius law and estimate the onset temperature by the deviation from the Arrhenius behaviour. We find that the onset temperature of the non-Arrhenius temperature dependence of τ_α is consistent with the crossover temperature of the exotic microphase ordering

[see Fig. R5a].

Following the reviewer's suggestion, we have characterised the partial self-intermediate scattering function of the three major components emerging in the exotic microphase ordering. Considering that there is a close correspondence between particle size and the coordination number, i.e., the smallest, medium, and largest particles correspond approximately to those with $z < 6$, $z = 6$, and $z > 6$ [see Fig. S2b of the Supplementary Information], we have calculated partial self-intermediate scattering functions for the smallest, medium, and largest particles (32%, 39%, and 29%, respectively; These fractions are chosen according to Fig. 2b at T_μ , and the results are found insensitive to minor variations of the values). This can help avoid particle identity fluctuation according to the coordination number z . The results of τ_α for the whole system, together with those for particles with different sizes, are shown in Fig. R5a. We find that particles with quite different sizes, and correspondingly different local structures, have very similar τ_α even at low temperatures. We further quantify the ratios of τ_α for particles with the smallest, medium, and largest sizes with respect to that for the whole system; and results are shown in Fig. R5b. Contrary to the simple expectation that τ_α of different components deviate more significantly from the average upon cooling, we find that the ratios for largest and smallest particles turn to approach one below T_μ . This observation strongly indicates that, with the development of the exotic microphase ordering, particles with different size are more involved in such correlated structures and therefore necessary to relax cooperatively. These results are included in the new Fig. 4 of the revised manuscript, which supports our conclusion that the development of the exotic microphase ordering is coupled with the change of dynamics.

We greatly thank the reviewer for this suggestion, which leads to the new finding of the structure-dynamics coupling.

Fig. R5: **a**, Temperature dependence of the structural relaxation time τ_α from standard MD simulations for the whole system and for particles with the smallest, medium, and largest sizes (32%, 39%, and 29%, respectively). High-temperature data points of τ_α for the whole system are fitted according to the Arrhenius law (solid line). **b**, Temperature dependence of

the ratio of τ_α for particles with the smallest, medium, and largest sizes with respect to that for the whole system. The dashed line corresponding to the ratio of 1 is a reference for eyes. The crossover temperature T_μ of the microphase ordering is indicated by the grey bar in both panels.

- The relevance of the section on the broken ergodicity via measurements of the heat-capacity with regards to the first part of the manuscript is unclear in the present form. The authors should explain the role of T_μ , which is indicated in Fig. 4 but not discussed in the context of microphase separation in neither the main text nor the S.I. Do the authors imply that T_μ is the point beyond which non-SWAP dynamics give the wrong result due to local sampling only as the figure seems to suggest? This would be wrong, as one could simulate for even shorter times with no swaps and have the curve fall off at $T > T_\mu$.

I suggest that the authors reformulate this section in a clearer and relevant way to the microphase separation they observe, or remove it entirely from the paper, as it pertains to a different topic: that of *averaging in near or non-ergodic systems*.

We thank the reviewer for this thoughtful comment. We agree with the reviewer that the crossover temperature T_μ is not that relevant in the discussion of broken ergodicity. So we have replotted Fig. 4 (Fig. 5 in the revised manuscript) without a particular indication of T_μ . On the other hand, the equilibration of the model system in this study is realised by swap MC at extremely low temperatures. It is, therefore, an intrinsic question of how to understand/characterise the thermodynamic properties of the low-temperature states when the ergodicity is effectively broken. The exotic microphase ordering at the lowest temperatures is expected to be the origin of significant energy barriers between statistically equivalent components in the phase space. This gives rise to the diverging timescale of physical relaxation and results in the broken ergodicity of this model glass former. Therefore, the section on broken ergodicity is an integral part of this work. Following the reviewer's suggestion, we have reformulated this section more clearly and relevantly.

- At the very end of the results, the authors write "Our findings thus evidence that, apart from dynamics, unphysical effects on other thermodynamic quantities may arise due to the unphysical particle movements included in SWAP". This is technically true in the case that a SWAP simulation were over a time-scale much longer than a typical experiment, thereby ergodically exploring a system that on experimental timescales is not ergodic. However, I believe the statement may be misleading, since the effective time-scales over which experiments are typically conducted are much closer than those effectively obtained through SWAP than through standard simulation methods. This means that the degree of ergodicity experienced in SWAP is practically much closer to the experiments than in standard simulations, which implies that not SWAP, but rather standard MD simulations should require a careful account for this ergodicity discrepancy.

We thank the reviewer for this very thoughtful comment. The concept of broken ergodicity in the glass transition problem is indeed subtle. We note that, at the lowest temperatures under study, the physical structure relaxation time is expected to diverge (below T_0 from the VFT law) or estimated to be far beyond the age of the

universe (see Ref. 28, using an Arrhenius fitting of low-temperature τ_α). This far exceeds any experimental time scale. In any practical sense, the system should be better understood and described as a solid. In such a situation, it is crucial to consider the broken ergodicity with special care to obtain physically relevant descriptions of the system. This point has been explicitly discussed in the seminal paper on the random-first-order transition theory of glass transition by Kirkpatrick, Thirumalai, & Wolynes [Phys. Rev. A 40, 1045 (1989)]. It was shown necessary to switch from the canonical ensemble (average over the whole phase space) to a restricted ensemble (first average within each glassy basin and then average over different basins) according to the broken ergodicity at low temperatures, to obtain the system's 'physical' free energy and entropy. This further leads to the **physically relevant** theoretical description of the glass transition problem.

We would emphasize that we fully agree that SWAP is an efficient algorithm in exploring the whole phase space and equilibrating the system. We also agree that with the nonlocal dynamic rule of SWAP, the system might be considered as liquid even the lowest temperatures. What we want to discuss in this section of the manuscript is the role of broken ergodicity in the **physically relevant** description of the low-temperature glassy state with realistic dynamics.

In addition, we point out that, different from the 3D polydisperse system [Ninarello, Berthier, & Coslovich, Phy. Rev. X 7, 021039 (2017)], where SWAP can only access the time scale over which the experiments are typically conducted, the speedup of SWAP in the current 2D model system is crazy. We speculate that this might also be related to the exotic microphase ordering revealed in this work. The exotic microphase ordering is expected to result in significant energy barriers between statistically equivalent components in the phase space. While structure relaxations are frozen out with physical dynamics in standard MD simulations, due to the continuous character of the ordering and large polydispersity, it is still possible to cross the considerable energy barriers by particle swaps, leading to the gradual evolution of the structure.

We have substantially revised this section with additional discussion to state these points clearly.

Minor comments and suggestions

- The authors use as quantifier of structure $S_6(k)$, which is a correlation function between density fluctuations which are weighed by the hexatic bond-orientational order parameter Ψ_6 . This should be appropriately referenced. It would be instructive to include a more in depth physical interpretation of this quantity. In particular this would help the reader have a better understanding of Fig. 2(d) where a peak emerges at low k and why/how it signals microphase separation. Additionally it would also help in the interpretation of the double peak observed between $10 < k < 15$. For the lowest temperature $T = 0.03$, the structure factor (Fig. S1) appears to show a premise of the double peak in Fig. 2(d) as well as a modest peak at low wave-vectors. Overall the section could be made much more convincing by also referring to, and showing in the main text, explicitly the partial structure factors $S_{z=6}(k)$ and $S_{z\neq 6}(k)$, where a very clear picture of the reported exotic

microphase separation is presented. Together with Fig. 2(c) the global picture of microphase separation would then be very convincing.

We thank the reviewer for this comment. Accordingly, in the revised manuscript, we have included a description of $S_6(k)$ as the Fourier transform of the two-point correlation function of Ψ_6 field. Overall, as evidenced by Fig. 1b, three major components emerge in the exotic microphase ordering: small and large particles with $z \neq 6$, which form the network-like structures, and medium particles with $z = 6$, which form patches in the pores of the network. Ψ_6 field can reflect this feature and, therefore, we choose to use it as the observable. We have added a reference [Ref. 27 in the revised manuscript] to illustrate the usage of $S_6(k)$. In addition, following the suggestion of the reviewer, we have included the partial structure factors $S_{z=6}(k)$ and $S_{z \neq 6}(k)$ as the new Fig. 3 in the revised manuscript.

The double peak observed between $10 < k < 15$ in $S_6(k)$ can be understood together with that observed in the partial structure factors $S_{z=6}(k)$. Since the particles with $z = 6$ form patches, which can be considered small pieces of a triangle lattice. The corresponding reciprocal lattice has basic vectors with amplitudes $2\pi/a$, $\sqrt{3} * 2\pi/a$, $4\pi/a$, ... The observed double peak is expected to be the vestige of the second and third peaks of the corresponding triangle lattice. Overall, those peaks become inconspicuous in the full static structure factor upon the global average, since different signals of the structure are mixed.

- The authors state that the microphase separation is between intermediately sized particles with 6-fold order and small/large particles with non-6-fold order. This relationship between the particle size and 6-fold order would be best illustrated by a figure that shows the relationship between D_i and Ψ^i .

We thank the reviewer for this valuable comment. We note that Fig. S2b of the Supplementary Information shows the average diameter of particles with different coordination numbers z at different temperatures. It gives the message that small, medium and large particles correspond approximately to particles with $z < 6$, $z = 6$, and $z > 6$. Following the reviewer's suggestion, we have plotted the relationship between particle diameter and Ψ_6 in Fig. R6, which clearly reveals three major species of particles with different sizes and 6-fold order. We have included this result as Fig. S3 in the revised Supplementary Information.

Fig. R6: A scatter plot of particle diameter σ verse Ψ_6 for a typical configuration at $T = 0.03$

- The authors claim that the microphase separation is between droplets of 6-fold order and a network of non-6-fold order. These droplets cause a small peak in the structure factor at $k \approx 3$, corresponding to lengthscales of about 2 particle diameters. This can be confirmed by Fig. 1b and 1c, where the yellow patches indeed have a characteristic size of around 2 particle sizes. Could the authors confirm that this is the right interpretation and if so, is 2 particle diameters large enough for a collection of particles to be called a droplet?

We thank the reviewer for this comment. Following the suggestion, we have changed the word “droplet” to “patch” in the revised manuscript.

- The authors claim that there are no ‘distinct thermodynamic signatures’ of the microphase separation. Could the authors be more precise on what thermodynamic quantities they have checked?

We thank the reviewer for this comment. What we have checked are the energy, pressure, and specific heat as a function of temperature. We have made this clear in the revised manuscript.

- In the context of the microphase separation observed, have the authors studied the dynamical features of particles with coordination number $z = 6$ and the other particles with $z \neq 6$? For example, the authors could study a partial intermediate scattering function or a partial non-Gaussian parameter. I believe this could greatly strengthen the main claim of the paper, as it would directly correlate the key features of the newly discovered structure with its dynamics.

We thank the reviewer for this comment. As the response to the above comment explains, we have characterised the partial self-intermediate scattering functions for

the smallest, medium, and largest particles. We are pleased that the new results further reveal the correlation between the newly discovered structural ordering and the dynamics, which significantly strengthens our paper's main claim

- In a recent paper [Scalliet, Guiselin, Berthier (2022)], the system studied in this work has been used to study a hypothesised facet of glassy dynamics known as 'dynamic facilitation'. Could the authors perhaps comment on the relevance of their results with respect to this work vis-a-vis dynamic heterogeneity at very low temperatures?

Since we have only studied the 2D system, we may try to understand the 'dynamic facilitation' observed in 2D from the perspective of the newly discovered structural ordering. At least at low temperatures, such structural ordering is significant, and the system might be considered an ordered state with certain fractions of defects. Then, similar to the case of a crystal with defects, the dynamics may initiate from those defects and spread in the form of defect migration. Figure 4 of that recent paper seems to be consistent with this scenario. Therefore, it is not clear how the observations of that paper carry to more generic glass formers. This would be an interesting direction for future studies.

- Finally, a minor comment, in many figures (2cd for instance), the authors display the legend in the form of a continuous color bar, while all the curves correspond to discrete values. The continuous color bar makes it very difficult to identify at which temperatures the simulations were performed exactly, thus hampering reproducibility. I suggest that they either label the different lines explicitly or put a list of studied temperatures in the Methods section.

We thank the reviewer for this comment. Accordingly, we have explicitly listed the studied temperatures in the caption of the figures.

We hope that the reviewer will find that all the concerns have now been adequately addressed and that our revised manuscript has been significantly strengthened and now deserves publication in *Nature Communications*.

REVIEWER COMMENTS

Reviewer #1 (Remarks to the Author):

I recommend the paper for publication in its present version. I still think that the discussion about the specific heat and the difference between SWAP and standard MC is misleading. The only conclusion that one can reach is that not reaching the Dulong and Petit law is an indication that there is no thermodynamic transition to an amorphous solid in this system. In any case, I think that the readers will be able to assess by themselves the validity of the interpretation put forward by the authors, which does not contain the main results anyway. In conclusion, I don't think one needs to delay further the publication because of this.

Reviewer #2 (Remarks to the Author):

The manuscript has improved, but unfortunately still contains presentation issues that make it unsuitable for publication. I hope the authors can address them in a last revision, as I think this is a study that deserves to be published eventually in Nature Communications.

1) The authors have changed their initial wording "exotic microphase separation" to "exotic microphase ordering". While this removes the possible confusion with actual microphase separation, it introduces a new ambiguity and also does not fully address the main point I raised.

The word "phase ordering" is often used in statistical physics to describe the growth of order (associated to the broken symmetry phase) during coarsening. The wording "microphase ordering" is confusing because it suggests to an out-of-equilibrium phenomenon, while the analysis here is at equilibrium. Moreover, it is not even clear what the "microphase" is. A phase (or microphase) should be identified through a well-defined order parameter, while here we can only see tendencies toward different kinds of coordination environments. I went through the revised paper, but I could not find a precise definition of what the microphase should be. In the absence of a thermodynamic signature this wording is misleading.

One should use a non-ambiguous wording for the interesting structural features of this model. My suggestion is to stick to the concept of "compositional order". I am impressed by the strong difference of the $P(\sigma_{NB}, \sigma)$ maps added in the SI: this 2d model has a much more marked compositional order than the 3d additive counterpart. Is this compositional order really "exotic"? Oxide glasses feature networks of nearly molecular units (e.g. tetrahedra in SiO_2) with precise chemical arrangements, not unlike what is seen here. But I agree that in systems with continuous polydispersity such features are somehow unusual and distinct from conventional scenarios (ex. fractionation). Thus, my advice is to replace "exotic microphase ordering" with "exotic compositional order" to remove all the above ambiguities.

One final point is that there is no evidence that this compositional order "interferes" with any glass transition (p.3: "Thus, the fundamental question remains whether other types of structural ordering may set in at low temperatures and interfere with the physics of supercooled liquids and glass transition.", in the abstract: "Therefore, the glass transition may be hybridised unexpectedly with the exotic microphase ordering"). By the same argument, the presence of stable structural units such as tetrahedra in an oxide glass could interfere with the glass transition (!). As long as there are no thermodynamic anomalies, compositional order does not interfere with anything.

2) The discussion of broken ergodicity is unfortunately still confusing and presents an erroneous interpretation of the swap MC results. There are several points that would have to

be discussed (well beyond what I can write in this report) so I will focus on a few key issues.

(a) First off, the goal of the swap MC simulations is to sample the canonical equilibrium distribution. If sampling is restricted to the non-crystalline portion of phase space, this gives access to ensemble averages representative of the (metastable) liquid down to very low temperature. Swap MC was used precisely to overcome the limitations of time averages with physical dynamics (in simulations or experiments). The sentence on p.11 "Our findings thus evidence that at least when considering the glass physics of realistic systems, apart from dynamics, the description of thermodynamic quantities like the specific heat could be altered by SMC due to the unphysical particle swaps" erroneously presents as problematic ("altering thermodynamic quantities") the very purpose of the method: measure equilibrium thermodynamic averages inaccessible in normal simulations or experiments. By the same argument, one would dismiss all the enhanced sampling methods that have been devised to bypass critical slowing down in the proximity of continuous phase transitions! The clumsy discussion in the last paragraphs at the end of p.11 appears to be based on a fundamental misconception about the purpose of these enhanced MC simulations. This also appears in the reply: "In contrast, the unphysical effects on thermodynamic quantities like the specific heat may arise due to the unphysical particle swaps included in SMC."

(b) Then we get to broken ergodicity. The authors discuss this aspect too superficially, both in the revised paper and reply.

In the reply, the authors consider a situation in which ergodicity is broken because of a broken symmetry: "In such a situation, if the canonical ensemble is employed to describe the system, i.e., allowing the magnetisation to flip in the infinitely long time limit, it is expected to predict a system with zero magnetisation on average. This is incorrect for the physical situation." and then forcefully put this together with the effectively broken ergodicity (on finite time scale) observed below the kinetic glass transition: "Due to the diverging time scale at deep supercooling, broken ergodicity would also affect the theoretical description of the glass transition phenomenon." Unless an ideal glass transition occurs, barriers remain finite in the thermodynamic limit in real glassy materials so these situations are different.

- The sketches of toy energy landscapes that appear in the reply and in the SI further demonstrate this confusion: the authors write "infinite energy barriers" (!) but of course the sketches only show finite ones. They must be removed.

- In the reply: "This has been explicitly discussed in the seminal paper on the random-first-order transition theory of glass transition by Kirkpatrick, Thirumalai, & Wolynes [Phys. Rev. A 40, 1045 (1989)], which shows the necessity of switching from the canonical ensemble to a restricted ensemble according to the broken ergodicity at low temperatures, for obtaining the system's 'physical' free energy and entropy. Our result of specific heat measurements is within these theoretical considerations."

This comment, which is reiterated on p.10, is erroneous. The issue of "physical" free energies and entropy is only problematic for mean-field models, for which barriers are infinite in the thermodynamic limit, as stated in Sec. II of the above paper:

"Note that the physical free energy, if the barriers are infinite, is F because the term, $-Tl$ in Eq. (2.6) is an entropy term which is a measure of parts of state space not probed in a finite amount of time."

The switch to the restricted average is only "necessary" in mean-field models, as again clearly stated in the cited reference, which also makes it clear that the restricted average is an approximation in finite-dimensional systems:

- "Technically Eq. (2.4) is correct (in the restricted ensemble) if the barriers between states diverge in the bulk limit. Physically it is a reasonable equation for a restricted time interval if the barriers are large but finite."

On p.11: "Even in this case, the physical relaxation time at the lowest temperature goes far beyond the age of the universe [28]. Then, in such low-temperature glassy states, the entire phase space should be decomposed into a collection of metabasins, referred to as "components", separated by significant energy barriers [40]. Physically, the system must be effectively trapped in one of the components within the observation time."

Once again, swap MC simulations are supposed to sample the equilibrium canonical distribution, not to be representative of a typical out-of-equilibrium measurement, which would depend on the observation time scale (no matter if it is the one of the universe).

- To conclude, for a finite-dimensional system, the partitioning of phase space into components (within which the system is effectively ergodic) is an approximation that can be used to describe the typical results of a measurements when the relaxation times largely exceed the observation times, i.e. when the system falls out of equilibrium. The hypothesis of confinement within a component is actually not trivial to test and a proper account of the continuous cooling typical of glassy materials requires extending the ideas of Palmer, see for instance the works by Mauro and collaborators on continuously broken ergodicity. The key point is that the broken ergodicity formalism allows one to describe the glass, whereas swap MC is concerned with the (metastable) equilibrium liquid. Hence I maintain that the discussion on broken ergodicity, as presented in the paper, is largely irrelevant to the purpose of this work and the corresponding discussion on p. 10 and 11 and in the supplement must be revised again, removing the erroneous implications pointed out above.

3) Like the authors, I am also surprised by the new Fig. 4: there is very little difference between structural relaxation times of particles of different sizes. I recommend to cross check these results once more and, if they are confirmed, to point out this is be a major difference between the 2d models and its 3d counterpart (for which the dynamics of the particles are quite dependent on size).

Reviewer #3 (Remarks to the Author):

I am pleased that the authors took all comments seriously and adapted their paper accordingly. In my opinion the paper has greatly improved from this.

Before publication, however, I have a few remaining questions and suggestions.

1. While the particle-size-resolved analysis is an important new addition to the paper, it actually raises a few new questions, since these results appear to be contradictory to existing literature (as the authors also mention). In particular, how is it possible that the difference in particle mobilities between large and small particles decreases as the temperature is lowered? Is this something that happens only in 2 dimensions? While the authors correct for Mermin-Wagner fluctuations by considering trajectories relative to the initial cage, given the fact that small and large particles have differently sized cages, are the authors sure that these effects cannot be at the root of their findings? Would the findings change if the bond-breaking correlation function (see eg the recent PRX by Scalliet, Guiselin and Berthier) is studied instead of the intermediate scattering function?

2. For completeness and reproducibility, could the authors please include a figure in the SI with the species-resolved intermediate scattering functions from which Figure 4 (main text) is

derived?

3. I still find the link between the “broken ergodicity” section and the rest of the paper unclear. If the authors do prefer to keep this section, I have a few final suggestions:

- Could the authors specifically indicate that the data labelled “ dE/dT ” in Fig. 5 is computed from the swap simulations and not from standard dynamics?
- It would make the section more compelling if the authors more clearly state the main message of this section. In particular, what static quantities can/cannot be measured from a simulation with swap dynamics? Could the authors comment on what this might imply for results from recent literature where static quantities are measured from SMC dynamics?

● **Response to Reviewer #1's Comments:**

I recommend the paper for publication in its present version. I still think that the discussion about the specific heat and the difference between SWAP and standard MC is misleading. The only conclusion that one can reach is that not reaching the Dulong and Petit law is an indication that there is no thermodynamic transition to an amorphous solid in this system. In any case, I think that the readers will be able to assess by themselves the validity of the interpretation put forward by the authors, which does not contain the main results anyway. In conclusion, I don't think one needs to delay further the publication because of this.

We thank the reviewer for kindly recommending the publication of our manuscript in *Nature Communications*.

We also appreciate the thoughtful comments on the part discussing the broken ergodicity of the previous manuscript. After careful reconsideration, we have decided to withdraw the discussion of broken ergodicity based on the specific heat measurement, partly because we realise that the ultimate explanation of this result relies on how to interpret the nature of the exotic structural ordering revealed in this work, more specifically, whether it can be considered as an ordinary glassy order or not. This is the fundamental question that arises from our findings, but answering it unambiguously is beyond the scope of the current work. Furthermore, the exotic compositional ordering already starts to appear in the temperature range where the system can be equilibrated without SMC. This implies that the ordering may not be induced by a peculiarity of SMC but may be an intrinsic feature of this particular liquid. Therefore, in the revised manuscript, we have focused only on the thermodynamic aspect that highlights the exotic nature of the compositional ordering.

We thank the reviewer again for assessing our manuscript carefully. We hope the reviewer finds that the revised manuscript is now suitable for publication in *Nature Communications*.

● Response to Reviewer #2's Comments:

The manuscript has improved, but unfortunately still contains presentation issues that make it unsuitable for publication. I hope the authors can address them in a last revision, as I think this is a study that deserves to be published eventually in Nature Communications.

We thank the reviewer for the appreciation of our work and for giving us the opportunity to revise the manuscript. The comments and suggestions provided by the reviewer were very thoughtful and constructive. We have seriously reconsidered them and substantially revised the manuscript accordingly. Below please find our detailed responses.

1) The authors have changed their initial wording "exotic microphase separation" to "exotic microphase ordering". While this removes the possible confusion with actual microphase separation, it introduces a new ambiguity and also does not fully address the main point I raised.

The word "phase ordering" is often used in statistical physics to describe the growth of order (associated to the broken symmetry phase) during coarsening. The wording "microphase ordering" is confusing because it suggests to an out-of-equilibrium phenomenon, while the analysis here is at equilibrium. Moreover, it is not even clear what the "microphase" is. A phase (or microphase) should be identified through a well-defined order parameter, while here we can only see tendencies toward different kinds of coordination environments. I went through the revised paper, but I could not find a precise definition of what the microphase should be. In the absence of a thermodynamic signature this wording is misleading.

One should use a non-ambiguous wording for the interesting structural features of this model. My suggestion is to stick to the concept of "compositional order". I am impressed by the strong difference of the $P(\sigma_{NB}, \sigma)$ maps added in the SI: this 2d model has a much more marked compositional order than the 3d additive counterpart. Is this compositional order really "exotic"? Oxide glasses feature networks of nearly molecular units (e.g. tetrahedra in SiO_2) with precise chemical arrangements, not unlike what is seen here. But I agree that in systems with continuous polydispersity such features are somehow unusual and distinct from conventional scenarios (ex. fractionation). Thus, my advice is to replace "exotic microphase ordering" with "exotic compositional order" to remove all the above ambiguities.

We thank the reviewer for the very constructive comment and suggestion. Accordingly, we have replaced "exotic microphase ordering" with "exotic compositional order".

One final point is that there is no evidence that this compositional order "interferes" with any glass transition (p.3: "Thus, the fundamental question remains whether other types of

structural ordering may set in at low temperatures and interfere with the physics of supercooled liquids and glass transition.", in the abstract: "Therefore, the glass transition may be hybridised unexpectedly with the exotic microphase ordering". By the same argument, the presence of stable structural units such as tetrahedra in an oxide glass could interfere with the glass transition (!). As long as there are no thermodynamic anomalies, compositional order does not interfere with anything.

We thank the reviewer for this comment. We do agree that without any thermodynamic signature, any ordering does not interfere with glass transition. With the sentence "Thus, the fundamental question remains whether other types of structural ordering may set in at low temperatures and interfere with the physics of supercooled liquids and glass transition" in the Introduction, we intended to ask the question: is there structural ordering other than crystallisation and phase separation (including demixing or fractionation) that does not belong to 'glassy order'? However, we admit that, before a clear assessment of the nature of the nonconventional structural order revealed in this work (i.e., is it acceptable as an ordinary glassy order or not?), it is not appropriate to use words which may imply that the "exotic compositional order" is not an ordinary glassy structural order. Accordingly, we have revised the manuscript to remove the ambiguity.

2) The discussion of broken ergodicity is unfortunately still confusing and presents an erroneous interpretation of the swap MC results. There are several points that would have to be discussed (well beyond what I can write in this report) so I will focus on a few key issues.

We thank the reviewer for the very thoughtful comments. Thanks to the comments, we have reconsidered the discussion of broken ergodicity seriously and decided to withdraw the discussion based on the specific heat measurement. This is partly because we realise that the ultimate explanation of this result relies on how to interpret the nature of the exotic compositional order revealed in this work, more specifically, whether it can be considered as an ordinary glassy order or not. This is the fundamental question that arises from our findings, but answering it unambiguously is beyond the scope of the current work. Furthermore, the exotic compositional ordering already starts to appear in the temperature range where the system can be equilibrated without SMC. This implies that the ordering may not be induced by a peculiarity of SMC but may be an intrinsic feature of this particular liquid. Therefore, in the revised manuscript, we have focused only on the thermodynamic aspect that highlights the exotic nature of the compositional ordering. We greatly thank the reviewer for leading us to realise this point and improve this part of the paper.

We might not need to respond to the following comments since we largely agree with the reviewer's opinions. Nevertheless, we will explain our considerations which were unclear in the previous manuscript and our replies.

(a) First off, the goal of the swap MC simulations is to sample the canonical equilibrium distribution. If sampling is restricted to the non-crystalline portion of phase space, this gives access to ensemble averages representative of the (metastable) liquid down to very low temperature. Swap MC was used precisely to overcome the limitations of time averages with physical dynamics (in simulations or experiments). The sentence on p.11 "Our findings thus evidence that at least when considering the glass physics of realistic systems, apart from dynamics, the description of thermodynamic quantities like the specific heat could be altered by SMC due to the unphysical particle swaps" erroneously presents as problematic ("altering thermodynamic quantities") the very purpose of the method: measure equilibrium thermodynamic averages inaccessible in normal simulations or experiments. By the same argument, one would dismiss all the enhanced sampling methods that have been devised to bypass critical slowing down in the proximity of continuous phase transitions! The clumsy discussion in the last paragraphs at the end of p.11 appears to be based on a fundamental misconception about the purpose of these enhanced MC simulations. This also appears in the reply: "In contrast, the unphysical effects on thermodynamic quantities like the specific heat may arise due to the unphysical particle swaps included in SMC."

We thank the reviewer for this comment. Actually, we fully agree with the reviewer that the swap MC method is very powerful in sampling the canonical equilibrium distribution. What we intended to discuss is not an issue of the swap MC method, but instead the glass transition problem.

As also mentioned by the reviewer, "If sampling is restricted to the non-crystalline portion of phase space, this gives access to ensemble averages representative of the (metastable) liquid down to very low temperature", what we are concerned is the possibility of a subtle structural order, other than crystal or phase separation, that does not belong to glassy order in the conventional sense (i.e., it is in the phase space outside of the liquid basin). Since glass transition is, by definition, concerned with only the metastable supercooled liquid basin, it would be a problem if the basin of such nonconventional structural order is unexpectedly visited.

Thanks to the reviewer's comment, we realise that our previous discussions are somehow biased towards the possibility that the exotic compositional order is not an ordinary glassy structural order. Only in such a situation, unphysical effects on thermodynamic quantities may arise due to the visit of different "components" by nonlocal particle swaps. We greatly thank the reviewer for leading us to realise and remove this bias.

(b) Then we get to broken ergodicity. The authors discuss this aspect too superficially, both in the revised paper and reply.

In the reply, the authors consider a situation in which ergodicity is broken because of a broken symmetry: "In such a situation, if the canonical ensemble is employed to describe the system, i.e., allowing the magnetisation to flip in the infinitely long time limit, it is

expected to predict a system with zero magnetisation on average. This is incorrect for the physical situation." and then forcefully put this together with the effectively broken ergodicity (on finite time scale) observed below the kinetic glass transition: "Due to the diverging time scale at deep supercooling, broken ergodicity would also affect the theoretical description of the glass transition phenomenon." Unless an ideal glass transition occurs, barriers remain finite in the thermodynamic limit in real glassy materials so these situations are different.

We thank the reviewer for this comment. We agree with the reviewer that the analogy is only validated between magnetisation and the glass transition in mean-field models and can only be an approximation in real glassy materials.

- The sketches of toy energy landscapes that appear in the reply and in the SI further demonstrate this confusion: the authors write "infinite energy barriers" (!) but of course the sketches only show finite ones. They must be removed.

We thank the reviewer for this comment. Together with the discussion of broken ergodicity, we have removed this part from the revised manuscript.

- In the reply: "This has been explicitly discussed in the seminal paper on the random-first-order transition theory of glass transition by Kirkpatrick, Thirumalai, & Wolynes [Phys. Rev. A 40, 1045 (1989)], which shows the necessity of switching from the canonical ensemble to a restricted ensemble according to the broken ergodicity at low temperatures, for obtaining the system's 'physical' free energy and entropy. Our result of specific heat measurements is within these theoretical considerations."

This comment, which is reiterated on p.10, is erroneous. The issue of "physical" free energies and entropy is only problematic for mean-field models, for which barriers are infinite in the thermodynamic limit, as stated in Sec. II of the above paper:

"Note that the physical free energy, if the barriers are infinite, is F because the term, $-Tl$ in Eq. (2.6) is an entropy term which is a measure of parts of state space not probed in a finite amount of time."

The switch to the restricted average is only "necessary" in mean-field models, as again clearly stated in the cited reference, which also makes it clear that the restricted average is an approximation in finite-dimensional systems:

- "Technically Eq. (2.4) is correct (in the restricted ensemble) if the barriers between states diverge in the bulk limit. Physically it is a reasonable equation for a restricted time interval if the barriers are large but finite."

We thank the reviewer for this comment. We agree with the reviewer that only for mean-field models, in which the barriers can be infinite, it is strictly validated that the switch

to the restricted ensemble is necessary. Therefore, in finite dimensions, the description of the glass transition within a restricted ensemble can only be an approximation.

We intended to say that, as stated in the paper [Kirkpatrick, Thirumalai, & Wolynes, Phys. Rev. A 40, 1045 (1989)], “Physically it is a reasonable equation for a restricted time interval if the barriers are large but finite”, one might expect it to be a reasonable and therefore effective description of the system (although it is not clear below which temperature the approximation is good). Our discussions in the previous reply were from this perspective.

On p.11: "Even in this case, the physical relaxation time at the lowest temperature goes far beyond the age of the universe [28]. Then, in such low-temperature glassy states, the entire phase space should be decomposed into a collection of metabasins, referred to as “components”, separated by significant energy barriers [40]. Physically, the system must be effectively trapped in one of the components within the observation time."

Once again, swap MC simulations are supposed to sample the equilibrium canonical distribution, not to be representative of a typical out-of-equilibrium measurement, which would depend on the observation time scale (no matter if it is the one of the universe).

We thank the reviewer for this comment. As stated in the above responses, we fully agree with the reviewer that the swap MC method is very powerful in sampling the canonical equilibrium distribution. The consideration behind our previous discussion was twofold. First, we are concerned about the possibility of a subtle structural order, other than crystal or phase separation, that does not belong to glassy order in the conventional sense. But we realise that our discussion is somehow biased towards the possibility that the exotic compositional order is not an ordinary glassy structural order. Second, the switch to a restricted ensemble may be a good approximation for a reasonable and effective description of the system with significant but finite barriers. In other words, it is about which ensemble, canonical or restricted, to be used in the effective description of the system (not an issue about the swap MC method). However, we agree that the discussion or proof of this point is beyond the scope of this work.

- To conclude, for a finite-dimensional system, the partitioning of phase space into components (within which the system is effectively ergodic) is an approximation that can be used to describe the typical results of a measurements when the relaxation times largely exceed the observation times, i.e. when the system falls out of equilibrium. The hypothesis of confinement within a component is actually not trivial to test and a proper account of the continuous cooling typical of glassy materials requires extending the ideas of Palmer, see for instance the works by Mauro and collaborators on continuously broken ergodicity. The key point is that the broken ergodicity formalism allows one to describe the glass, whereas swap MC is concerned with the (metastable) equilibrium liquid. Hence I maintain that the discussion on broken ergodicity, as presented in the paper, is largely irrelevant to the purpose of this work and the corresponding discussion on p. 10 and 11 and in the

supplement must be revised again, removing the erroneous implications pointed out above.

We thank the reviewer for the thoughtful comments and kind suggestions. As explained above, our previous discussions were indeed somehow biased toward the possibility that the exotic compositional order is not an ordinary glassy structural order. Therefore, we have removed the discussion of broken ergodicity from the main text and the supplementary information.

3) Like the authors, I am also surprised by the new Fig. 4: there is very little difference between structural relaxation times of particles of different sizes. I recommend to cross check these results once more and, if they are confirmed, to point out this is be a major difference between the 2d models and its 3d counterpart (for which the dynamics of the particles are quite dependent on size).

We thank the reviewer for the suggestions. We have carefully checked and confirmed these results. In the revised manuscript, we have included a discussion on the difference between the 2d model and its 3d counterpart with a new reference [new Ref. 36, a very recent work on the 3d model by Pihlajamaa, Laudicina, & Janssen, arXiv:2302.09549 (2023)].

Finally, we thank the reviewer again for carefully reading our manuscript and for providing very constructive comments and criticisms. We believe that the revisions following the reviewer's suggestions have increased the solidity and clarity of our manuscript.

We hope the reviewer finds that all presentation issues have been adequately addressed and that our revised manuscript is now suitable for publication in *Nature Communications*.

● Response to Reviewer #3's Comments:

I am pleased that the authors took all comments seriously and adapted their paper accordingly. In my opinion the paper has greatly improved from this.

We are pleased that the reviewer thought the manuscript was greatly improved. We also thank the reviewer for further questions and suggestions. Below, please find our detailed responses.

Before publication, however, I have a few remaining questions and suggestions.

1. While the particle-size-resolved analysis is an important new addition to the paper, it actually raises a few new questions, since these results appear to be contradictory to existing literature (as the authors also mention). In particular, how is it possible that the difference in particle mobilities between large and small particles decreases as the temperature is lowered? Is this something that happens only in 2 dimensions? While the authors correct for Mermin-Wagner fluctuations by considering trajectories relative to the initial cage, given the fact that small and large particles have differently sized cages, are the authors sure that these effects cannot be at the root of their findings? Would the findings change if the bond-breaking correlation function (see eg the recent PRX by Scalliet, Guiselin and Berthier) is studied instead of the intermediate scattering function?

We thank the reviewer for the thoughtful questions. Indeed, the dynamic behaviour of the current 2D model system appears unusual. We have carefully checked again and confirmed these results. It differs from the corresponding 3D model system with the same interacting potential and particle size distribution, for which the particle mobility strongly depends on the size [see new Ref. 36 in the revised manuscript; a very recent work by Pihlajamaa, Laudicina, & Janssen, arXiv:2302.09549 (2023)]. We expect that such a peculiar structural relaxation behaviour originates from the nonconventional structural ordering revealed in this work. The network-like structure involves large and small particles, whereas the small patches of medium-size particles are in its pores. Therefore, the relaxation of such correlated structures is expected to be cooperative, and the particles with different sizes are necessary to relax together. Thus, with the growth of the exotic compositional order upon cooling, the size-dependent mobility difference becomes smaller. We have included a discussion in the revised manuscript on these points.

The dynamics in 2d has been characterised using both the relative motion and the bond-breaking correlation function in previous studies [see Shiba, et al., PRL117, 245701 (2016); Viveka, et al., PNAS 114, 1850 (2017)]. Binary systems with a size ratio of 1:1.4, 2.53:3.38, and 1.1:2.6 are used. The comparison of results suggests that the two methods give essentially the same information (which is reasonable since both measure the relative dynamics between a central particle and its neighbours). In our system, the average size ratio of small and large particles is 0.78:1.28~1:1.6, which is not very

different from these studies. Therefore, we may not expect an essential change in the findings if the bond-breaking correlation function is used.

We do agree with the reviewer that the dynamic feature of the current 2d model is unusual and raises new interesting questions. This is worthwhile to be explored in more detail in a follow-up study.

2. For completeness and reproducibility, could the authors please include a figure in the SI with the species-resolved intermediate scattering functions from which Figure 4 (main text) is derived?

We thank the reviewer for this suggestion. Accordingly, we have included a figure for the species-resolved intermediate scattering functions as new Fig. S6 in the revised SI.

3. I still find the link between the “broken ergodicity” section and the rest of the paper unclear. If the authors do prefer to keep this section, I have a few final suggestions:

- Could the authors specifically indicate that the data labelled “dE/dT” in Fig. 5 is computed from the swap simulations and not from standard dynamics?

- It would make the section more compelling if the authors more clearly state the main message of this section. In particular, what static quantities can/cannot be measured from a simulation with swap dynamics? Could the authors comment on what this might imply for results from recent literature where static quantities are measured from SMC dynamics?

We thank the reviewer for this comment and further suggestions. After seriously reconsidering the key points of our work, we have decided to withdraw the discussion of broken ergodicity based on the specific heat measurement. This is partly because we realise that the ultimate explanation of this result relies on how to interpret the nature of the exotic structural ordering revealed in this work, more specifically, whether it can be considered as an ordinary glassy order or not. This is the fundamental question that arises from our findings, but answering it unambiguously is beyond the scope of the current work. Furthermore, the exotic compositional ordering already starts to appear in the temperature range where the system can be equilibrated without SMC. This implies that the ordering may not be induced by a peculiarity of SMC but may be an intrinsic feature of this particular liquid. Therefore, in the revised manuscript, we have focused only on the thermodynamic aspect that highlights the exotic nature of the compositional ordering.

Following the advice, we have specifically indicated that the data labelled “dE/dT” in Fig. 5 is computed from the swap simulations and not from standard dynamics in the revised manuscript.

We hope the reviewer finds that the concerns have been adequately addressed and that our revised manuscript has been improved and now deserves publication in *Nature Communications*.

REVIEWERS' COMMENTS

Reviewer #2 (Remarks to the Author):

The authors have addressed all the concerns raised in my report and I now recommend the paper for publication.

Reviewer #3 (Remarks to the Author):

I thank the authors for the final revisions they have made, and am happy to recommend the current version for publication as is.